



# The carbon sink in China as seen from GOSAT with a regional inversion system based on CMAQ and EnSRF

Xingxia Kou[1], Zhen Peng[2]*, Meigen Zhang[3, 4], Fei Hu[3,4], Xiao Han[3,4], Ziming Li[5], Lili Lei[2,6]

[1]Institute of Urban Meteorology, China Meteorological Administration, Beijing, China

[2]School of Atmospheric Sciences, Nanjing University, Nanjing, China

[3]State Key Laboratory of Atmospheric Boundary Layer Physics and Atmospheric Chemistry, Institute of Atmospheric Physics, Chinese Academy of Sciences, Beijing, China

[4]University of Chinese Academy of Sciences, Beijing, China

[5]Beijing Meteorological Observatory, Beijing, China

[6]Key Laboratory of Mesoscale Severe Weather, Ministry of Education, Nanjing University, Nanjing, China

*Correspondence to: Z. Peng, pengzhen@nju.edu.cn*

**Abstract.** Top-down inversions of China's terrestrial carbon sink are known to be uncertain because of errors related to the relatively coarse resolution of global transport models and the sparseness of *in situ* observations. Taking advantage of regional chemistry transport models for mesoscale simulation and spaceborne sensors for spatial coverage, the Greenhouse Gases Observing Satellite (GOSAT) column-mean dry mole fraction of carbon dioxide ($XCO_2$) retrievals were introduced in the Models-3 Community Multi-scale Air Quality (CMAQ) and Ensemble Square Root Filter (EnSRF)-based regional inversion system to constrain China's biosphere sink at a spatiotemporal resolution of 64 km and 1 h. In general, the annual, monthly and daily variation in biosphere flux was reliably delivered, attributable to the novel flux forecast model, reasonable CMAQ background simulation, well-designed observational operator, and joint data assimilation scheme (JDAS) of $CO_2$ concentrations and fluxes. The size of the assimilated biosphere sink in China was −0.47 PgC yr$^{-1}$, which was consistent with most global estimates (i.e., −0.27 to −0.68 PgC yr$^{-1}$), indicating that the regional inversion system was sufficient to robustly constrain the control vectors. Furthermore, the seasonal patterns were recalibrated well, with a growing season that shifted earlier in the year over central and south China. Moreover, the provincial-scale biosphere flux was re-estimated, and the difference between the *a posteriori* and *a priori* flux ranged from −7.03 TgC yr$^{-1}$ in Heilongjiang to 2.95 TgC yr$^{-1}$ in Shandong. Additionally, better performance of the *a posteriori* flux in contrast to the *a priori* flux was proven when the



simulation was fitted to independent observations, indicating improved results in JDAS. This study serves as a basis for future regional- and urban-scale top-down carbon assimilation.

## 1 Introduction

In the context of human-induced climate change, the Paris Agreement charts the course for the world to transition to a green way of development and outlines the minimum steps to be taken to protect the Earth, which requires all countries to make significant commitments to stabilize atmospheric greenhouse gas concentrations and keep the global average temperature to well under a 2℃ rise (UNFCCC 2015). Therewith, a growing number of countries and regions are pledging to achieve net-zero emissions in the second half of this century; for instance, Austria by 2040, Sweden by 2045, the European Union by 2050, and China by 2060. Hence, there has been an increasing demand from policymakers and the scientific community in general for accurate knowledge of $CO_2$ emissions from anthropogenic sources (so that the targeted reductions are effective) and from biospheric uptake (so that natural reservoirs remain stable) (Ciais et al., 2015; Pinty et al., 2017; Friedlingstein, et al., 2020; Deng et al., 2022). In 2019, the Intergovernmental Panel on Climate Change (IPCC) published a refined methodology report as an update to its 2006 guidelines with the aim to complement them with a bottom-up, transparent framework and highlight the Monitoring and Verification Support (MVS) capacity using independent atmospheric measurements (IPCC, 2019). A great deal of effort has been devoted in recent decades to developing and applying atmospheric $CO_2$ inversions to constrain global- and regional-scale $CO_2$ fluxes (Enting et al., 1995; Thompson and Stohl, 2014; Broquet et al., 2011, Peters, et a., 2007; Tian et al., 2014; Kou et al., 2017; Kountouris et al., 2018). Most of these inversions are informed by ground-based observations and global chemistry transport models (CTMs), which is far from sufficient to support the abovementioned needs. Despite the development of surface observation networks with highly accurate continuous data, such as ICOS (the Integrated Carbon Observation System) in Europe, the global distribution of ground-based $CO_2$ measurements remains rather sparse and inhomogeneous. Consequently, errors are introduced, CTMs lack accuracy, and assimilation frameworks deliver inconsistent regional flux estimates obtained using state-of-the-art global inversions from national up to continental scales (Monteil et al., 2020; Piao et al., 2022; Schuh et al., 2022).



Spaceborne sensors, designed specifically to retrieve atmospheric concentrations with unprecedented
      spatial coverage, have in recent years begun to improve the current understanding of greenhouse gases
      and the associated $CO_2$ emissions' MVS capacity. At present, there are several operational $CO_2$
      observation satellites in orbit, including Japan's Greenhouse Gases Observing Satellite (GOSAT; Kuze
      et al., 2009), GOSAT-2 (Glumb et al., 2014), the US Orbiting Carbon Observatory 2 (OCO-2, Eldering

et al., 2017a, 2017b), OCO-3 (Eldering et al., 2019), and China's TanSat (Liu et al., 2018; Yang et al.,
      2018). It is recognized that satellite retrievals of shortwave infrared radiation, despite their uncertainty,
      are sufficient to reliably capture the seasonal variability of $XCO_2$ (column-mean dry mole fraction of
      carbon dioxide), as a first-order question in constraining inversion models (Lindqvist et al., 2015; Li et
      al., 2017). Furthermore, several centers and universities routinely assimilate GOSAT $XCO_2$ data into

models to estimate terrestrial ecosystem carbon exchange, including Japan's National Institute for
      Environment Studies (NIES), the United States' National Aeronautics and Space Administration
      (NASA), France's Laboratoire des Sciences du Climat et de I'Environnement, the Netherland's
      Institute for Space Research, the UK's University of Edinburgh, Canada's University of Toronto, and
      China's Nanjing University. As an example, the NIES GOSAT Project provides a Level 4 $CO_2$ data

product, and the monthly regional $CO_2$ flux estimates for the period 2009–2013, based on $XCO_2$
      retrievals and NIES' global atmospheric tracer transport model with Bayes' theorem, are publicly
      available (Maksyutov et al., 2013; Takagi et al., 2014). Furthermore, NASA's Carbon Flux Monitoring
      System is another recent top-down global inversion system configured with 4DVar and GEOS-Chem
      (Goddard Earth Observing System with Chemistry) and concurrently assimilates $XCO_2$ from GOSAT

and OCO-2. It has released the longest available terrestrial flux estimates (from 2010–2018) on
      self-consistent global and regional scales and has planned future updates of the dataset on an annual
      basis (Liu et al., 2021). In addition, the University of Edinburgh has simultaneously produced a
      five-year $CH_4$ and $CO_2$ flux estimate for 2010–2014 directly from GOSAT retrievals of $XCH_4$:$XCO_2$ by
      using GEOS-Chem and an ensemble Kalman filter (EnKF) (Feng et al., 2017). Moreover, the Global

Carbon Assimilation System has been upgraded by Nanjing University to assimilate the GOSAT $XCO_2$
      retrievals from 2010–2015 with the Ensemble Square Root Filter (EnSRF) algorithm and the Model for
      Ozone and Related Chemical Tracers, version 4 (Jiang et al., 2021; 2022). Overall, the top-down $CO_2$
      biosphere flux datasets inverted from satellite data suggest an improved flux estimation compared with



the large uncertainties in process-based terrestrial biosphere model estimates (Byrne et al., 2019;

Chevallier et al., 2019; Chen et al., 2021). Deng et al. (2016) and Wang et al. (2018) further highlighted

the importance of improved observational coverage to better quantify the latitudinal distribution of

terrestrial fluxes by combining GOSAT observations over land and ocean. Also, the sensitivity of

observations from GOSAT and OCO-2 to optimized $CO_2$ fluxes has been examined using GEOS-Chem,

indicating that GOSAT offers greater sensitivity in Northern Hemisphere spring and summer (Byrne et

al., 2017; Wang et al., 2019).

Nevertheless, a GOSAT $CO_2$ global inversion intercomparison experiment involving eight research

groups found that, as expected, the most robust flux estimates were obtained at large scales and quickly

diverged at subcontinental scales; and the inversions primarily involved uncertainties in their global

CTMs, satellite retrievals, *a priori* fluxes, and inversion frameworks (Chevallier et al., 2015;

Houweling et al., 2015; Fu et al., 2021). Generally, the assimilated $CO_2$ flux (i.e., the analytical field) is

a weighted average of background information and observations, which depends on the correlation

coefficient between simulated concentrations of the observation and the state variable (i.e., $CO_2$ flux).

In particular, considering the transport errors introduced by global CTMs, the reliability of the regional

fluxes inferred from GOSAT retrievals remains a topic of ongoing discussion (Reuter, et al., 2017; He

et al., 2022). Consequently, if we can configure a reasonable simulation of the background $CO_2$

concentration compared with the coarse spatiotemporal resolution of the global scale, then the flux

constrained by observations can be estimated more precisely at national and city scales. The step up in

inversion resolution and accuracy calls for new developments in shifting from global to regional

inversions.

However, regional CTMs, with their advantages in resolving fine-scale $CO_2$ concentrations, are rarely

used in satellite carbon data assimilations, even though multimodel comparisons have reported large

uncertainties introduced by global CTMs in estimating the carbon sink of China's biosphere (Wang et

al., 2021; Piao et al., 2022; Schuh et al., 2022; Wang et al., 2022). Notably, the use of regional CTMs in

$CO_2$ research is becoming more commonplace. For instance, Huang et al. (2014) demonstrated the

importance of regional CTM performance to data assimilation and suggested it is possible to improve

the simulation accuracy of the synoptic-scale variation in atmospheric $CO_2$ by utilizing the EnKF



framework and CMAQ (Multi-scale Air Quality Modeling System). Zhang et al. (2021) assimilated

OCO-2 retrievals with WRF-Chem/DART (Weather Research and Forecasting model coupled with Chemistry/Data Assimilation Research Testbed) to improve the estimation of $CO_2$ concentrations. For regional $CO_2$ inversions inferred from surface stations, towers, and aircraft flights, several studies have in recent years relied on the FLEXPART Lagrangian model, CHIMERE (France's multi-scale CTM), WRF-Chem, and CMAQ to estimate not only urban $CO_2$ emissions in megacities (e.g., Los Angeles,

Paris, Indianpolis), but also terrestrial ecosystem exchange over Europe, North America, and East Asia (Brioude et al., 2013; Staufer et al 2016; Lauvaux et al 2016; Thompson et al., 2016; Kou et al., 2017; Zheng et al., 2018; Monteil et al., 2021). Moreover, the potential use of CMAQ and EnKF in regional $CO_2$ inversions with GOSAT retrievals has been explored by Peng et al. (2015) with observing system simulation experiments. Pillai et al. (2016) also concluded that satellite missions such as CarbonSat

(Carbon Monitoring Satellite) have high potential to obtain city-scale $CO_2$ emissions by using a high-resolution modeling framework.

Previous studies have highlighted that the simultaneous assimilation of concentrations and fluxes as state variables can help reduce the uncertainty of both the initial $CO_2$ fields and the fluxes (Tian et al.,

2014; Peng et al., 2015; Kou et al., 2017). Recently, Peng et al. (2017, 2018, 2020) improved air quality forecasts and emission estimates over China by developing a novel flux forecast model with the EnSRF-based Joint Data Assimilation Framework (JDAS), so that the extended model can construct ensembles of both concentration and flux at the hourly scale. As an extension to this work, JDAS was further developed towards a high-resolution inversion of $CO_2$ fluxes based on CMAQ and EnSRF with

real-time GOSAT observations over China from 1 January 2016 to 31 December 2016, which holds an advantage over global models in terms of the $CO_2$ background information and inversion scheme. To the best of our knowledge, this is the most up to date estimates of China's biosphere flux informed by a regional CTM and satellite observations. It should prove to be of considerable value, particularly under the framework of the Paris Agreement, which requires high spatiotemporal resolution inversions of

$CO_2$ flux for carbon accounting at national scales.

In this paper, we focus on the development of top-down estimates constrained by GOSAT retrievals and CMAQ. Using this unique regional inversion technique, we address the following questions:


1. On what scales can regional CTMs facilitate the inversion of GOSAT observations compared with global inversions?

2. What is the difference between flux inversions from spaceborne retrievals and ground-based observations? Are they inconsistent?

## 2 Model, System and Data

### 2.1 CMAQ regional transport model

The atmospheric transport and the signature of sources and sinks in $CO_2$ concentrations were simulated using a regional CTM, i.e., CMAQ, which was originally developed by the US Environmental Protection Agency to model multiple air quality issues over a variety of scales, and has been updated for passive tracers, as in Kou et al. (2013) with a 1–64 km horizontal resolution capability. The CMAQ

regional modeling system has already been used in several regional studies and has shown promising performance in capturing the fine-scale spatiotemporal variability of $CO_2$ mixing ratios (e.g., Kou et al., 2013; 2015; Liu et al., 2013; Huang et al., 2014; Li et al., 2017). The CMAQ configuration used here was a domain of 6720 km $\times$ 5504 km with 64 $\times$ 64 km$^2$ fixed grid cells centered at 35 °N and 116 °E in a rotated polar stereographic map projection. This domain, having 105 (west–east) $\times$ 86 (south–north)

grid points, covered the whole of mainland China and its surrounding regions (Fig. 1). The model has 15 vertical layers unequally spaced from the ground to approximately 23 km, half of which are concentrated in the lowest 2 km to improve the simulation of the atmospheric boundary layer. In addition, RAMS (Regional Atmospheric Modeling System) provides the three-dimensional meteorological fields, with the lowest seven layers being the same as those in CMAQ. The time step of

the CMAQ output is 1 h.

The initial and lateral boundary meteorological fields, sea surface temperatures, and initial soil conditions were prescribed by European Centre for Medium-Range Weather Forecasts reanalysis data with a spatial resolution of $1°\times 1°$ and 6-hourly temporal intervals (Zhang et al., 2002). As the real

initial and lateral boundary atmospheric $CO_2$ concentrations, the global 4D $CO_2$ data were created using the optimized surface fluxes and simulated atmospheric transport of CarbonTracker, version CT2019B,





from the National Oceanic and Atmospheric Administration (NOAA), with a spatial resolution of $3° \times 2°$, 25 vertical levels, and a temporal resolution of 3 h, which represent the optimum estimate of the distribution of atmospheric $CO_2$ (Jacobson et al., 2020). In addition, the *a priori* biosphere and ocean

fluxes used for simulations within the CMAQ domain were also derived from the CT2019B optimized fluxes at a 3-h intervals, but with a spatial resolution of $1° \times 1°$. The anthropogenic $CO_2$ emission fluxes were based on the Multi-resolution Emissions Inventory for China, version 1.3, and the Regional Emissions Inventory in Asia, version 3.2, with monthly gridded data at a resolution of $0.25° \times 0.25°$ (Zheng et al., 2018; Kurokawa et al., 2020). The Global Fire Emissions Database, version 4.1s, with

monthly gridded data at a resolution of $0.25° \times 0.25°$, was applied to provide the biomass burning emissions (van der Werf et al., 2017). The abovementioned four individual $CO_2$ fluxes (i.e., biosphere, fossil fuels, fire, and ocean) were spatially interpolated to the CMAQ grid, conserving the total mass of emissions. In each EnSRF analysis step, CMAQ integrated and generated a 3D $CO_2$ concentration ensemble derived by the $N$ ensemble fluxes with perturbed $CO_2$ initial and boundary conditions.

**2.2 JDAS $CO_2$ inversion framework**

The inverse optimization updates EnSRF, originated from NOAA's operational EnKF system (https://dtcenter.ucar.edu/com-GSI/users/docs/users_guide/GSIUserGuide_v3.7.pdf), to assimilate the GOSAT observations in order to optimize the surface biosphere $CO_2$ fluxes. The EnSRF algorithm has been extended to simultaneously assimilate multiple chemical initial conditions and emissions with the

*in situ* measurements of their atmospheric observations, and produce one of the latest Chinese reanalysis datasets of atmospheric composition as well as an updated emissions inventory (Peng et al. 2017, 2018, 2020; Kou et al., 2021). In the present study, the initial $CO_2$ concentrations and fluxes were also designed to be concurrently assimilated within the JDAS framework, which indicates that both the $CO_2$ concentrations and fluxes were regarded as state variables (i.e., $\boldsymbol{x} = \begin{bmatrix} \boldsymbol{C}, \boldsymbol{E} \end{bmatrix}^T$), and helpful

observational information employed in the current assimilation cycle could be efficiently capitalized upon in the next assimilation cycle with reduced uncertainty in the initial $CO_2$ conditions.

$CO_2$ flux was treated as the model input, with the result that ensemble samples of fluxes could not be prepared by the CTM's forward forecasting. Consequently, besides the application of the CMAQ model

to generate ensemble $CO_2$ concentrations in JDAS, the forecast model also consisted of a novel flux





forecast model, which was designed to generate the background $CO_2$ flux ensembles $\boldsymbol{E}^{f}_{i,t+1}$, where $i =$ 1,…, $N$ refers to the $i$th ensemble member at time $t$ (Equation 1). The superscripts $a$, $f$ and $p$ denote "assimilation", "forecast" and "$a$ $priori$", respectively. First, following Peng et al. (2020), the $a$ $priori$ flux ensemble $\boldsymbol{E}^{p}_{i,t+1}$ is created by using the ensemble CMAQ forecast $CO_2$ concentration $\boldsymbol{C}^{f}_{i,t}$

forced by the $\boldsymbol{E}^{f}_{i,t}$, where $\overline{\boldsymbol{C}^{f}_{t}} = \dfrac{1}{N}\sum_{i=1}^{N}\boldsymbol{C}^{f}_{i,t}$ stands for the ensemble mean of $\boldsymbol{C}^{f}_{i,t}$ and $\boldsymbol{E}^{p}_{t+1}$

refers to the $a$ $priori$ flux. The covariance inflation factor $\beta$ is further used to keep the ensemble spread of the $CO_2$ concentration scaling factor $\boldsymbol{\kappa}_{i,t}$. The ensemble mean of $\boldsymbol{\kappa}_{i,t}$ can be expressed as

$\overline{\boldsymbol{\kappa}_{t}} = \dfrac{1}{N}\sum_{i=1}^{N}\boldsymbol{C}^{f}_{i,t}\Big/\overline{\boldsymbol{C}^{f}_{i}} = 1$ . Next, in the second part of Equation (1), the ensemble mean of

$\overline{\boldsymbol{E}^{f}_{t+1}} = \dfrac{1}{M}\left(\sum_{j=M-1}^{1}\overline{\boldsymbol{E}^{a}_{t-24\times j}} + \boldsymbol{E}^{p}_{t+1}\right)$ is determined by the assimilated $CO_2$ flux at the same time on

each day from the previous assimilation cycles among these $M-1$ days (i.e., $\overline{\boldsymbol{E}^{a}_{t-24\times(M-1)}}$ ,

$\overline{\boldsymbol{E}^{a}_{t-24\times(M-2)}}$ ,…, and $\overline{\boldsymbol{E}^{a}_{t-24\times1}}$, $j = M-1, M-2, \cdots, 1$ ) and the $a$ $priori$ $CO_2$ flux $\boldsymbol{E}^{p}_{t+1}$. $M$ refers to the length of the smoothing window, which was chosen as 4 days. This design follows Peters et al. (2007), in which the useful observational information from the previous assimilation cycle was made beneficial to the next assimilation cycle via a smoothing operator but was further modified to cooperate

with the diurnal variation in $CO_2$ biosphere flux. Then, $\overline{\boldsymbol{E}^{f}_{t+1}}$ was used to recenter $\overline{\boldsymbol{E}^{p}_{t+1}}$. In contrast to previous flux models without diurnal variation, this new flux model is advantageous insofar as it facilitates the development of assimilation between regional CTM forecasts and observations at the hourly scale, so as to achieve high-resolution inversion. Thus, the background of the joint vector, $\boldsymbol{x}^{f} = \left[\boldsymbol{C}^{f}, \boldsymbol{E}^{f}\right]^{T}$ , can be prepared. Furthermore, the associated analyzed state vector,

$\boldsymbol{x}^{a} = \left[\boldsymbol{C}^{a}, \boldsymbol{E}^{a}\right]^{T}$ , can be updated by applying the EnSRF constrained by GOSAT retrievals:

$$\boldsymbol{E}^{f}_{i,t+1} = \boldsymbol{E}^{p}_{i,t+1} + \left(\overline{\boldsymbol{E}^{f}_{t+1}} - \boldsymbol{E}^{p}_{t+1}\right)$$
$$= \beta\left(\dfrac{\boldsymbol{C}^{f}_{i,t+1}}{\boldsymbol{C}^{f}_{t+1}} - \overline{\boldsymbol{\kappa}_{t}}\right)\boldsymbol{E}^{p}_{t+1} + \dfrac{1}{M}\left(\sum_{j=M-1}^{1}\overline{\boldsymbol{E}^{a}_{t-24\times j}} + \boldsymbol{E}^{p}_{t+1}\right)$$

(1)



In this study, by developing an observational operator, EnSRF was further extended to be able to assimilate the GOSAT $XCO_2$ retrievals. The simulated $CO_2$ concentration profiles were mapped into

the satellite retrieval levels and then vertically integrated based on the satellite averaging kernel according to the following equation:

$$XCO_2^f = XCO_2^p + \sum_{k=1}^{N_{lev}} \left\{ \left[ \left( y_k^f - y_k^p \right) A_k \right] h_k (1-w) \right\}, \qquad (2)$$

where the subscript $k$ represents the retrieval level, $XCO_2^p$ denotes the *a priori* $XCO_2$ for retrieval,

$y_k^p$ is the *a priori* $CO_2$ profile for retrieval, $A_k$ stands for the satellite column-averaged kernel, $h_k$

is a pressure weighting function, and $y_k^f$ denotes the CMAQ-simulated $CO_2$ profile interpolated into

the corresponding retrieval levels. As in Equation 1, the superscripts *f* and *p* refer to "forecast" and "*a priori*" in Equation 2. Moreover, $w$ denotes the RAMS water mole fraction, which was used to map from the $CO_2$ concentrations to the dry mole fraction, as suggested by Feng et al. (2009). A brief description of the GOSAT retrievals and operations before assimilation is given in Section 2.3.


The basic configuration of the JDAS $CO_2$ inversion settings followed previous studies. For instance, the ensemble size *N* was set to 50 to sustain the balance between computational cost and ensemble performance. The horizontal covariance localization radius was chosen as 1280 km to localize the observation's impact and ameliorate the spurious long-range correlations between state variables and

observations caused by the limited number of ensemble members. Moreover, the covariance inflation factor *β* was set to 80 to preserve the ensemble spread. In this study, the assimilation window was set to 24 h, and hour-by-hour assimilation was adopted in the novel flux forecast model and fine-scale CMAQ background simulation. Hereafter, AN denotes the analysis fields $\left[ C^a, E^a \right]$ and BG denotes the model's first guess background fields $\left[ C^f, E^f \right]$ in the assimilation.

**2.3 GOSAT $XCO_2$ retrievals**

GOSAT, launched by the Japan Aerospace Exploration Agency in January 2009, was designed to make near-global greenhouse gas measurements in a sun-synchronous orbit. It covers the whole globe in 3 d and has a sounding footprint of approximately 10.5 km. In this study, we assimilated GOSAT $XCO_2$





retrievals from NASA's Atmospheric $CO_2$ Observations from Space Level 2 standard data products

(version ACOS_L2_Lite_FP.9r; data available at

https://oco2.gesdisc.eosdis.nasa.gov/data/GOSAT_TANSO_Level2/). The $XCO_2$ data from Lite

products were bias-corrected (Wunch et al. 2017; O'Dell et al. 2018). Typically, Level 2 Lite products

contain 10–200 useful soundings per orbit, noting that more than 50% of the spectral data were not

processed during retrieval because they did not pass the first cloud screening pre-processing step.


Before being applied in the JDAS inversion system, the GOSAT retrievals were operated in three steps.

First, the $XCO_2$ retrievals were filtered with the "outcome_flag" parameter, which indicates the

retrieval quality and are provided along with the ACOS product. Only data retrievals tagged with

"RetrievalResults/outcome_flag =1" were selected, particularly where soundings converged. Second, to

achieve the most extensive spatial coverage with the assurance of using the best quality data available,

a thinning strategy was used when multiple observations appeared in the same model grid at the same

hour on each day after interpolation of the model's horizontal coordinates. Only retrievals with the

minimum value of uncertainty, i.e., "RetrievalResults/xco$_2$_uncert", were selected, which represented a

higher quality of retrieval data. According to the statistics listed in Table 2, the total number of thinned

$XCO_2$ values in 2016 was 19267, with the highest coverage in January (~2300) and lowest coverage in

July (~730). Third, the difference between the observation and first guess of the model (denoted as

$o-b$) was further tested, based upon which, if the difference between the $XCO_2$ retrieval and the

CMAQ background–simulated $XCO_2$ was greater than a certain threshold value (±5.00 ppm), the

retrieval was further excluded from the JDAS inversion to maintain stability in the assimilation. The

total number of assimilated $XCO_2$ values in 2016 reached 15264 (i.e., 79.22% of the thinned amount),

with the monthly ratio of "assimilated-to-thinned" ranging from 74.19% (in August) to 98.91% (in

July). It should be noted that the maximum median $XCO_2$ uncertainty occurred in July (0.99 ppm) and

the minimum in December (0.64), indicating a better quality of $XCO_2$ retrievals in winter and less

stable retrievals in summer. The scenario of $|o-b| > 5.00$ (i.e., the absolute value of $o-b$) was mostly

found near the boundary of the model domain.

Non-assimilated $XCO_2$ observations were used for verification purposes after another process of

repeated sifting, whose steps were as follows: (1) observations were marked with "outcome_flag = 1",



which selected the $XCO_2$ values that passed the internal quality check; (2) $XCO_2$ values with the

minimum "xco$_2$_uncert" in the same model grid and at the same hour were excluded, which filtered out

all of the assimilated $XCO_2$; (3) outliers were precluded if the absolute bias between the $XCO_2$ of the

analysis concentration field and the corresponding $XCO_2$ measurements was larger than 5.00 ppm (i.e.,

the same threshold as in the assimilation). In general, the total number of $XCO_2$ retrievals used for

validation in 2016 was 14660, ranging from ~2300 in January to ~730 in July (Table 2).

**2.4 Experimental design and evaluation method**

Following previous GOSAT inversion work (Maksyutov et al., 2013; Feng et al., 2017; Wang et al.,

2019; Liu et al., 2021; Jiang et al., 2022), in this study, the natural flux (i.e., biosphere–atmosphere

exchange and ocean–atmosphere exchange) were assimilated, while the fossil-fuel and

biomass-burning fluxes were kept unchanged. This design, in which the natural fluxes were a subset of

the state vector, further allowed us to focus on investigating the uncertainty of China's carbon sink,

since the uncertainty in prescribed biomass-burning and fossil-fuel emissions are minor compared to

that of the biosphere fluxes in the model domain (van der Werf et al., 2017; Zheng et al., 2018;

Kurokawa et al., 2020). Fully reconciling the differences between bottom-up and inversion-estimated

fossil-fuel emissions is outside the scope of this work and is therefore not discussed any further in this

paper. Consequently, the selected $XCO_2$ observations were assimilated hourly to adjust the initial $CO_2$

concentrations and fluxes. The ensemble assimilation was performed for the period 0000 UTC 25

December 2015 to 2300 UTC 31 December 2016 using the perturbed initial conditions, boundary

conditions, and natural fluxes by adding Gaussian random noise with a standard deviation of 5% and

10% of the corresponding variables, respectively. The first 7 days were set as spin-up, and results for

the period 1 January to 31 December 2016 are discussed and validated in detail in this paper.

Then, additionally, to assess the quality of the inversion results, two sets of forward simulations were

performed throughout the year of 2016. One set of experiments was forced by the optimized *a

posteriori* fluxes (denoted as FC), and the other was forced by the prescribed *a priori* fluxes as a

control experiment (denoted as CTRL). Both forward runs used the same initial and boundary

concentrations from the CT2019B product. Generally, it is hard to validate the optimized flux, because

comparison with *in situ* flux measurements is high-risk on account of the discrepancy in scales between





fluxes assimilated in the model grid and eddy-covariance measurements over a very large uniform underlying surface. Therefore, this traditional approach was adopted as a compromise to assess whether

the *a posteriori* fluxes would enable improvements in the fit to independent (i.e., non-assimilated) observed $CO_2$ concentrations.

## 3 Results and Discussion

### 3.1 Performance of observational and analysis increments

We begin by assessing the GOSAT observational performance in $CO_2$ concentration and flux joint

assimilation. In $CO_2$ inversion, usually, the $o - b$ is denoted as "innovations", and the analysis concentration and flux are obtained by adding the innovations to the model first guess with the weights that are determined based on the estimated statistical error covariance of the forecast and observations. Fig. 1 demonstrates the distribution of $XCO_2$ observation increments and $CO_2$ flux analysis increments over the model domain, including January (Figs. 1a and b), July (Figs. 1c and d) and the whole year

(Figs. 1e and f). Also, detailed statistical information on the assimilated $XCO_2$ is given in Table 2. The number of observations corresponding to each grid point in 2016 in the domain is approximately between 0 and 60, covering every province of China. Using January and July as the reference, predominant seasonal variation in the spatial coverage of $XCO_2$ occurs, with the most abundance in winter and the least in summer (Fig. 1), which is primarily associated with the screening depending

upon the extent of cloud coverage and aerosol filtering (Wunch et al. 2017). The available $XCO_2$ data amount for JDAS decreases from 1788 in January, to 1870 in February, to 734 in June, and to 728 in July, representing an approximate 61% reduction in the year-round monthly comparison (Table 2). In particular, most of the available $XCO_2$ in July appears in the north and central region of China, but the south and northwest tend to be blank. The $XCO_2$ innovation range is usually between −3 and 3 ppm in

the corresponding model grid, with a monthly mean value between −0.12 and −0.96 ppm over the model domain. As expected, the observational increments show an ability to depict the fine-scale features with strong spatial heterogeneity whilst in general retaining the large-scale spatial patterns, which can be attributed to the CMAQ simulation performance in differentiating the nuances of anthropogenic and natural conditions. In contrast, Fu et al. (2022) found that the results of a global

model (i.e., GEOS-Chem) tended to be generally lower than GOSAT's $XCO_2$ in China from the





weighted ensemble mean of various terrestrial models with a mean bias of about 2 ppm in winter, while Lei et al. (2014) found GEOS-Chem simulations tended to produce higher values than GOSAT (by 5.8 ppm) in China during summer. As shown in Table 2, the correlation between the CMAQ background simulation and the GOSAT assimilation is highest in July (0.80) and lowest in May (0.16). In addition,

both the mean absolute error (MAE) and root-mean-square error (RMSE) exhibit a maximum in July (1.99 and 2.41, respectively) and a minimum in April and September (MAE: 1.76 and 1.76 ppm; RMSE: 2.18 and 2.15 ppm), indicating that the point-by-point uncertainty is larger in summer and lower in spring and autumn, which is consistent with the seasonal performance from previous model studies (Li et al., 2017). This discrepancy of the seasonal scale could be partly due to the uncertainties

in the spatial and temporal variations of the biosphere flux estimation and fossil-fuel inventories. Generally, the shortwave near-infrared detectors mounted on GOSAT have been testified as being more sensitive to near-surface $CO_2$ changes (Buchwitz et al., 2013; O'Dell et al. 2018), which further demonstrates the potential to reduce the uncertainty of surface $CO_2$ flux estimates by assimilating GOSAT column concentration values.


The pattern of $CO_2$ flux analysis increments (i.e., AN–FC flux) demonstrated in Fig. 1 preserves features from innovations and certifies that GOSAT $XCO_2$ is effectively absorbed in JDAS. GOSAT retrievals were found to display impacts within a certain range near the observation points after entering the assimilation system. The monthly flux analysis increments vary from −0.2 to 0.1 μmol m$^{-2}$

s$^{-1}$ in January, and from −1.0 to 1.0 μmol m$^{-2}$ s$^{-1}$ in July, respectively. The higher variation in monthly flux analysis increments for July than those for January indicates that the uncertainties of forecast flux in summer are larger than those of the variation in winter. In this study, the biosphere flux first-guess fields were derived from the novel flux forecast model by taking the *a priori* flux, the analysis flux from the previous assimilation cycle, and the forecast concentration as independent variables (Equation

1), which is a great help in assisting with improving the background information and initial perturbation for ensemble forecasting. On the other hand, the EnSRF analysis increments depend not only on the innovations, but also on how well the Kalman gain matrix computes the contribution weighting factors based on the time-dependent forecast error covariance. Considering the peculiarities of atmospheric $CO_2$, such as its long atmospheric lifetime, long-range transport, high background

concentrations, and strong biosphere–atmosphere exchanges, there are both wide-ranging overall





increases (e.g., −0.01 to 0.1 over central China) and decreases (e.g., −0.2 to −0.01 over South China) and small-scale adjustment taking place in 2016 (Fig. 1f). In general, the flux analysis increments are reasonably and effectively calculated, which may be attributable to the novel flux forecast model, the favorable CMAQ forecast concentration, the representative observation increments, and the

well-designed assimilation framework.

### 3.2 Size of the annual carbon sink in China

Before presenting the *a posteriori* biosphere fluxes in China from JDAS, the total annual carbon sink in previous research along with our study are summarized (Table 1). The aim was mainly to check that all methods—for instance, inventories, ecosystem process models, and atmospheric inversions—actually

improve the carbon sink comparability, but also to check the reliability and credibility of the inversions. Based on national ecosystem inventory data, China's terrestrial carbon sink increased from −0.18 PgC yr$^{-1}$ in the 1980s to −0.33 PgC yr$^{-1}$ in the 2000s owing to forest area expansion and afforestation during recent years (Piao et al, 2009; Jiang et al., 2016; Wang et al., 2022). Meanwhile, the results from several ecosystem process-based models display a carbon sink ranging from −0.13 to −0.22 PgC yr$^{-1}$

during 1980–2010, achieved by assessing the effect of changes in climate and $CO_2$ (Piao et al, 2009; He et al., 2019). In addition, according to the flux gap between top-down and bottom-up estimations mentioned above, those atmospheric inversion results with lateral flux adjustment are also reported in Table 1 (italic and shaded parts). The lateral fluxes include the carbon exchange between the land and atmosphere in non-$CO_2$ forms as well as the imported wood and crop products, and a recent estimate of

the lateral flux for China is −0.14 PgC yr$^{-1}$ (Wang et al., 2022). The terrestrial carbon sink in 2016 after correcting for lateral fluxes amounts to approximately −0.33 PgC yr$^{-1}$ (i.e., −0.47 + 0.14 = −0.33), constrained by the GOSAT $XCO_2$ in the JDAS inversion system. This result is consistent with inventory-based and ecosystem model–based estimations within the recognized interannual variability. Correspondingly, we also provide a corrected carbon sink estimate of −0.54 PgC yr$^{-1}$ (i.e., −0.68 + 0.14

= −0.54) inferred from *in situ* $CO_2$ data provided by JDAS, which is the optimal mathematical solution under the current sparse observational coverage with daytime photosynthetic uptake, and likely leads to a slight overestimation to some extent.

Furthermore, as well as results for China's annual carbon sink, Table 1 also provides an overview of



most of the well-known inversion modeling systems, configurations of inversions, atmospheric transport models, spatiotemporal resolutions, and observations. In general, most research into the inversion of China's carbon sink has commonly used global transport models. The limited resolution and distribution of observations are deemed to lead to large uncertainties in inversion in small regions, especially at national scales (Scrowell et al., 2019; Monteil et al., 2020; Piao et al., 2022). The

resolution-related performance of transport models tends to magnify the uncertainty in China's carbon sink estimates, which can be attributed to the significant bias in representing atmospheric $CO_2$ concentrations with a coarse model resolution. For example, either *in situ* $CO_2$ or GOSAT $XCO_2$ constrained flux (i.e., −1.11 and −0.83 PgC yr$^{-1}$) demonstrates much higher sink estimates from GEOS-Chem-based inversion with a $4\,°\times5\,°$ horizontal resolution. Excluding the outliers, most global

inversions report a carbon sink in China of −0.27 to −0.56 PgC yr$^{-1}$ from *in situ* $CO_2$, and −0.34 to −0.68 PgC yr$^{-1}$ from satellite retrievals. In contrast, our estimates constrained by analogous observation (−0.68 and −0.47 PgC yr$^{-1}$ from *in situ* $CO_2$ and GOSAT, respectively) agree reasonably well with the previous estimates mentioned above, implying that the underlying regional transport model (i.e., CMAQ) is reliable in presenting robust local signals. Overall, the good agreement between JDAS

ground-based and satellite-based estimates, together with the comparable results from previous studies, suggests that the JDAS inversion configuration is sufficient to robustly constrain the control vector, and that the limited observations are effectively absorbed at the regional scale. This reinforces our confidence in analyzing and interpreting the optimized fluxes in terms of spatial variability over China.

### 3.3 Spatial variability of optimized fluxes

As can be seen in Fig. 2a, the annual horizontal distribution patterns of biosphere flux show significant spatial heterogeneity and fairly large gradients in most areas. Fig. 2b further illustrates annual differences between *a priori* and *a posteriori* fluxes over the model domain. Compared to the prescribed *a priori* biosphere flux, not only large-scale vegetation adjustments but also small-scale conditions can be detected throughout the year after assimilating atmospheric observations under the

UNFCCC's MVS framework (Fig. 2b). Although China's total carbon sink of *a posteriori* fluxes (−0.47 PgC yr$^{-1}$) are approximately equal to the *a priori* fluxes (−0.43 PgC yr$^{-1}$), the spatial distribution has been modified through assimilation. Generally, the *a priori* biosphere fluxes are overestimated (~0.1–0.3 μmole m$^{-2}$ s$^{-1}$) in the north (dominated by forest, grassland and cropland) and south (dominated by





forest and grassland) of China, while they are underestimated (~0.1–0.5 µmole m$^{-2}$ s$^{-1}$) primarily in

central China where there is a large area of cropland (He et al., 2022). This change in flux pattern needs

to be further assessed and discussed. The good response of the vegetation condition to the *a posteriori*

results provides a strong foundation for a meaningful interpretation of biosphere fluxes.

Figs. 2c–f show the seasonal spatial differences before and after assimilation, taking January, April,

July and October as representatives of winter, spring, summer and autumn. The monthly averages were

calculated from the daily averages based on hourly outputs. The seasonal spatial variation of biosphere

flux is considerably affected by the seasonal growth and decay of terrestrial ecosystems, which is

mainly driven by the variation in temperature, precipitation, photosynthetically active solar radiation,

and other meteorological factors (Fu et al., 2022). Accordingly, the difference between the analysis and

*a priori* flux tends to be larger in July (Fig. 2e; approximately −1.0 to 1.0 µmole m$^{-2}$ s$^{-1}$), lower in

April and October, and lowest in January, which indicates a larger uncertainty in biosphere flux

estimates in the growing season. This is consistent with the findings of previous studies (Jiang et al.,

2016; Chen et al, 2021). Nevertheless, summer is also the season with the largest percentage of satellite

data rejection and retrieval uncertainty, making it a tough test still for inversion systems. As a result,

JDAS maintains a robust and stable capability with better use of observational information throughout

the whole year, owing to the joint assimilation of $CO_2$ concentrations and fluxes helping to fully utilize

and absorb observations as well as reduce the uncertainties in initial concentrations fields. Moreover, it

should be noted that an obvious underestimation of *a priori* flux (approximately 0.1–0.5 µmole m$^{-2}$ s$^{-1}$)

occurs in the northern, central and southern vegetation growth regions, where there are several of

China's key ecological engineering construction areas, which will be further discussed later in detail.

On the other hand, the central part of China, dominated by cropland, shows relatively larger *a

posteriori* flux in winter and smaller *a posteriori* flux in summer and autumn, in contrast with the *a

priori* flux constrained by the limited background observation sites (Zhang et al., 2014; Jacobson et al.,

2020). Satellites, with their better spatial coverage, as well as regional transport models, with their

improved stability, can help in assessing the real conditions of local terrestrial ecosystems with

complex conditions, such as over central China. Additionally, compared with the weekly temporal

resolution of global inversion, the hourly observational increments as well as the hourly first-guess

fields in this study hold some advantage in evaluating the monthly variations of fluxes. As expected,



some distinguishing features are thus demonstrated in the assimilated fluxes, such as the carbon sources

in parts of central, eastern and southwest China, which is more consistent with the underlying surface

situation. In this way, the JDAS inversion system has the potential to depict the characteristics of

biosphere flux well.

Next, we analyze the monthly and annual fluxes in five large regions—west, north, central, south, and

mainland China (denoted by the red frame in Fig. 2a)—to evaluate the effectiveness of the regional

inversion in subcontinental-scale flux variation as well as to contrast with the previous inversion

analysis over China (Fig. 3). The flux forecast model that includes a smoothing operator with diurnal

variation provides reasonable background flux information. Given the representative background and

observation information, the seasonality patterns are reproduced well by the JDAS assimilation, with

larger annual sinks relative to the *a priori* ones and a growing season that is shifted earlier in the year

over central and south China. This indicates that the regional carbon assimilation system is calibrated

well and performs reliably. As shown in Fig. 3, there is an evident difference in the *a posteriori* annual

carbon sink magnitude in these regions, gradually decreasing in the north (e.g., forest, grassland and

cropland), south (e.g., forest and grassland), west (e.g., grassland and tundra), and central region (e.g.,

cropland) in turn, which is consistent with the primary corresponding ecosystem types, while the *a*

*priori* sink of the west tends to be larger than that of the south. Using the north as a reference, the

annual carbon sink of the *a priori* estimates for the north, south, west and central regions are 1.00, 0.57,

0.62 and 0.44, respectively, while those of the *a posteriori* estimates are 1.00, 0.62, 0.56 and 0.38. On

the other hand, the *a priori* and *a posteriori* amplitudes of the seasonal variation [i.e., the difference

between the maximum and minimum monthly estimates, as defined in Scrowell et al. (2016)] range

from 374.33/333.74, 87.01/80.41, 120.33/113.98, 82.34/88.00 to 413.17/389.48 TgC month$^{-1}$ in north,

south, west, central and mainland China, respectively. The decreased annual sink and increased

seasonal variability in central China deduced by the *a posteriori* flux with satellite observations may in

fact reflect the atmospheric $CO_2$ fixed by cropland vegetation, where ~60% of the area is cropland with

relative few *in situ* observations used for constraining the *a priori* flux (Piao et al., 2009, 2022).

Moreover, for daily flux estimation, the day-to-day variability demonstrated by *a posteriori* fluxes is

substantially smaller than that of the *a priori* estimation (sub-graph in the left-hand panel of Fig. 3).

The drastic fluctuation in the daily variation of *a priori* fluxes has been modified by observational



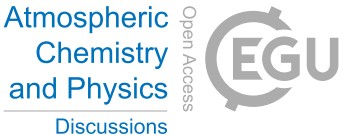

constraints, which appears more realistic than that of the *a priori* estimates. This implies the potential

for regional inversion in interpreting underlying processes in large regions such as China where the

ecosystems and climate are quite varied.

Nevertheless, achieving robust and reliable flux signals at smaller regional scales is quite demanding

and rather challenging, because of the limited observations and low accuracy of transport models as

well as the *a priori* information. In this paper, we further try to investigate the condition of the regional

biosphere carbon sink over several of China's key ecological areas (denoted by the blue frame in Fig.

2a)—for example, Daxing'anling (DX), the Loess Plateau (HT), the Qinling Mountains (QL), the rocky

desert in Guangxi (SM), Mount Wuyi (WY), and Xishuangbanna (XS). These regions are characterized

by their unique vegetation and climatic conditions. Generally, the duration of the carbon sink extends

gradually from north to south, such as four months in DX, five months in HT, and seven months in SM

and XS, due to the seasonal growth and decay of biosphere ecosystems, which is principally

determined by meteorological conditions including solar radiation, temperature and precipitation. In

particular, the *a priori* and *a posteriori* seasonal amplitudes amount to 43.64/39.56, 24.03/23.39,

35.73/37.96, 29.36/31.80, 2.70/3.64 and 7.93/7.04 TgC month$^{-1}$ in DX, HT, QL, SM, WY and XS,

respectively. The region of DX is characterized by abundant forest and far more satellite retrievals to

constrain fluxes, with annual *a priori* and *a posteriori* carbon sinks of −25.13/−29.64 TgC yr$^{-1}$.

Favorable meteorological conditions [e.g., precipitation in the growing season being 20% higher than

that in 2015 (China Climate Bulletin 2016)] have also been reported, which further supports the

improved ecological quality, indicating JDAS's potential in tracking biosphere $CO_2$ fluxes from space.

Compared to *a priori* fluxes, relatively stronger *a posteriori* sinks are also found in QL (−60.05/−62.53

TgC yr$^{-1}$), SM (−62.10/−71.27 TgC yr$^{-1}$), WY (0.36/−2.19 TgC yr$^{-1}$) and XS (−10.12/−10.79 TgC yr$^{-1}$),

which is consistent with the improved ecological conditions due to ecological engineering construction

as well as generally favorable climatic conditions. The XS region is unique and worthy of attention in

contrast to the other regions not only because it shows different seasonality in its release of $CO_2$ to the

atmosphere in summer and removal of $CO_2$ from the atmosphere in other seasons, but also because of

the large transport model errors that are included in the model–data mismatch error involved in

previous inversion studies (Wang et al., 2020; He et al., 2022; Schuh et al., 2022; Wang et al., 2022). As

can be seen in Fig. 4, JDAS demonstrates potential in reproducing a reasonable biosphere flux


dominated by complex underlying conditions, with a reliable and robust CMAQ performance in providing first-guess concentration fields. Thus, the abovementioned spatial variations of *a posteriori* fluxes might unlock some of the potential local signals in areas where regional transport models are more reliable and observations are plentiful.

### 3.4 Provincial patterns of optimized fluxes in China

In this section, we investigate the provincial patterns of biosphere flux. Based on the gridded *a* 525 *posterior* flux dataset, we first assess the annual $CO_2$ biosphere sink levels in 31 provinces in mainland China (Taiwan, Hong Kong, Macao and Shanghai are not discussed because of the insufficient grid resolution). Fig. 5 shows the *a priori*, *a posteriori* annual biosphere flux estimations and their differences (in units of µmole m$^{-2}$ s$^{-1}$) on the provincial scale over mainland China. At this scale, the inversion fluxes are associated with regional differences partly controlled by the *a priori* flux and the 530 atmospheric measurements. Both the *a priori* and *a posteriori* fluxes indicate the strongest carbon sink intensity per unit area (> 0.3 µmole m$^{-2}$ s$^{-1}$) being in Shaanxi, Guangxi and Guizhou, but the *a priori* fluxes produce an underestimation in Shanxi (~0.01–0.05 µmole m$^{-2}$ s$^{-1}$) and overestimations in Guangxi and Guizhou (~0.1–0.2 µmole m$^{-2}$ s$^{-1}$), respectively. Next, the second strongest carbon sink intensity (0.2–0.3 µmole m$^{-2}$ s$^{-1}$) is commonly seen in Shaanxi, Sichuan, Chongqing and Hubei, 535 whereas a comparatively low level of carbon sink intensity appears in Xinjiang, Liaoning, Anhui and Yunnan, at approximately 0.05–0.1 µmole m$^{-2}$ s$^{-1}$, as well as in Tibet and Fujian, at 0.01–0.05 µmole m$^{-2}$ s$^{-1}$. Furthermore, some provinces with neutral (i.e., close to 0), source or sink statuses are re-evaluated by the GOSAT constrained fluxes (Figs. 5a and b). For instance, the *a posteriori* flux in Ningxia is −0.01–0.01 µmole m$^{-2}$ s$^{-1}$, while the *a priori* flux displays a weak carbon sink of −0.01 to 540 −0.05 µmole m$^{-2}$ s$^{-1}$, due to the complexity in the estimation related to the grassland and cropland land surfaces in this province. On the contrary, the *a priori* fluxes in Fujian and Jiangsu are close to 0, but we find a carbon sink ranging from approximately −0.01 to −0.05 µmole m$^{-2}$ s$^{-1}$ and a carbon source from 0.05 to 0.1 µmole m$^{-2}$ s$^{-1}$, respectively. For Liaoning, the *a priori* fluxes are characterized by $CO_2$ sources (0.01–0.05 µmole m$^{-2}$ s$^{-1}$), while the assimilated fluxes with satellite measurements are slightly 545 adjusted to a carbon sink (−0.05–0.1 µmole m$^{-2}$ s$^{-1}$). In general, (1) widespread underestimation of the *a priori* flux (0.01–0.1 µmole m$^{-2}$ s$^{-1}$) is found in central China, which is dominated by cropland and where dense satellite retrievals are accordingly available; (2) overestimates are distribute in the



northeast and south of China over a considerable spatial extent and should be modified; and (3) smaller

changes between *a posteriori* and *a priori* estimates are primarily located in the west of China, which

tends to agree with the $XCO_2$ o − b pattern.

Lastly, the sizes of the provincial biosphere fluxes are summarized and sorted quantitatively in Fig. 6.

The maximum and minimum provincial biosphere flux sizes are in Inner Mongolia (*a posteriori*: −

53.65 TgC yr$^{-1}$; *a priori*: −53.41 TgC yr$^{-1}$) and Shandong (*a posteriori*: 5.99 TgC yr$^{-1}$; *a priori*: 3.05

TgC yr$^{-1}$), respectively. Moreover, satellites observations can facilitate the evaluation of biosphere flux

in combination with atmospheric inversions. The difference between the *a posteriori* and *a priori*

provincial flux ranges from −7.03 TgC yr$^{-1}$ in Heilongjiang to 2.95 TgC yr$^{-1}$ in Shandong, with an

underestimation greater than 2.00 TgC yr$^{-1}$ appearing in Shandong (2.95), Jiangsu (2.31) and Hebei

(2.25), and an overestimation greater than 5.00 TgC yr$^{-1}$ appearing in Heilongjiang (7.03), Liaoning

(5.68), Yunnan (5.59) and Guangxi (5.10). On the other hand, a smaller percentage of modification

between the *a posteriori* and *a priori* flux [i.e. (*a posteriori* − *a priori*) / *a priori* ×100% in absolute

value] arises in Xinjiang (0.28%), Inner Mongolia (0.46%), Tibet (1.10%), Qinghai (2.45%), Gansu

(3.21%), Shaanxi (3.50%), Sichuan (4.34%) and Shanxi (4.65%), indicating a lower level of

uncertainty in these larger carbon-sink provinces. Nevertheless, an increased percentage of

modification in provincial flux appears in Jiangsu (*a posteriori*: 2.29 TgC yr$^{-1}$; *a priori*: −0.02 TgC

yr$^{-1}$), Liaoning (*a posteriori*: −4.27 TgC yr$^{-1}$; *a priori*: 1.40 TgC yr$^{-1}$), Fujian (*a posteriori*: −1.15 TgC

yr; *a priori*: 0.29 TgC yr$^{-1}$), and Shandong (already listed above). As discussed earlier, all provinces in

China differ in both their terrestrial vegetation and anthropogenic activity. The abovementioned

magnitude of uncertainty between *a posteriori* and *a priori* estimates is closely related to the degree of

human activity intervention. Several factors could account for the provincial spatial distribution

constrained from GOSAT; for instance, the increased precipitation along with the strong El Niño in

2016, the levels of reforestation and afforestation, and the reductions in biofuels in rural areas bringing

about a shrubland carbon sink.

**3.5 Evaluation of a posteriori fluxes against independent data**

In this section, we further assess the performance of the *a posteriori* $CO_2$ fluxes by comparing the

CTRL, FC and AN results. The monthly and annual statistics were computed from the hourly outputs


from the assimilation, simulation and GOSAT retrievals. Table 2 demonstrates (as expected) that the concentration from the analysis fields (AN) performs best when fitted to the independent $XCO_2$ observations. Generally, the simulation with *a posteriori* fluxes (i.e., FC) shows improvements, with

decreased RMSE and MAE as well as an increased correlation coefficient, when compared to the *a priori* flux simulation (CTRL) using the non-assimilated $XCO_2$ for validation. It is notable that the column-averaged satellite signals have limited capacity in facilitating the tropospheric variation in $CO_2$ concentration, and thus the response to changes in the simulated concentration signal is weak, but improvements are still apparent. For instance, the annual RMSE, MAE and correlation coefficient for

AN are 2.34 ppm, 1.93 ppm and 0.73; for FC, they are 2.63 ppm, 2.02 ppm and 0.66; and for CTRL, they are 2.65 ppm, 2.03 ppm and 0.66, respectively. Additionally, the AN, FC and CTRL biases from independent observations were further calculated (Table 3). The outliers in CTRL have been effectively amended. When FC is compared with the CTRL results, the frequency of bias in [−4, 4] increases by 0.25%, in [−3, 3] by 0.36%, in [−2, 2] by 0.32%, and in [−1, 1] by 0.14%. Furthermore, the error

standard deviation decreases from 2.63 ppm in CTRL to 2.61 ppm in FC and to 2.27 ppm in AN.

Moreover, the annual-averaged horizontal distributions of $CO_2$ concentration (unit: ppm) near the surface in 2016 are also presented (Fig. 7). Fig. 7a displays the surface $CO_2$ concentration analysis fields, from which it can be seen that the high $CO_2$ concentrations are mainly distributed over regions

with intense human activities. Thus, the AN can be used as a closer representation of the real condition, and the much-refined description in the $CO_2$ analysis concentration fields allows for a more detailed characterization of the spatiotemporal distribution of $CO_2$ concentration and can further facilitate an interpretation of satellite data in a regional context over China. As shown in Figs. 7b and c, compared to the CTRL fields, the FC fields tend to be considerably closer to the AN fields, suggesting that the *a*

*posteriori* fluxes are calibrated well and perform acceptably. Furthermore, Fig. 7d shows the year-round statistic of $XCO_2$ error reduction [defined as $(1 - \delta_{FC} / \delta_{CTRL}) \times 100\%)$], as well as the amounts of independent observations, where $\delta_{FC}$ represents the FC $XCO_2$ error standard deviation and $\delta_{CTRL}$ the CTRL $XCO_2$ error standard deviation. The region of 8 °–57 °N and 105 °–120 °E is used as a reference because there is a relatively larger difference between the *a priori* and *a posteriori* fields,

including the concentration as well as flux. In general, the error reduction is primarily found to be positive and ranges from approximately 0.80% to 32.13% with a median of 5.65% and mean of 7.23%.



This zonal evaluation further verifies the improvement in the *a posteriori* flux compared to the *a priori* flux.

**4 Summary and Outlook**

Top-down estimations of carbon budgets have been included in the UNFCCC's MVS framework. At present, most carbon sink inversions in China utilize a global transport model with relatively coarse resolution. Characterized by large heterogeneity in its biospheric spatiotemporal distribution, the transport model error, as well as the sparseness of *in situ* observations, leads to large uncertainties in the assimilation of carbon flux in China. In this study, a regional high-resolution inversion model

(JDAS) was used, which has been extended to incorporate GOSAT constraints, along with a joint assimilation of $CO_2$ flux and concentration at high spatial (64 km) and temporal (1 h) resolution. The annual, monthly and daily variation in biosphere flux was reproduced reasonably well, which was attributable to the novel flux forecast model with diurnal variation, the reliable CMAQ background simulation, carefully chosen $XCO_2$ retrievals, and the well-designed EnSRF assimilation configuration.


The size of the biosphere carbon sink in China amounted to $-0.47$ PgC yr$^{-1}$ with JDAS by GOSAT constraints, which is consistent with previous global estimates (i.e., $-0.27$ to $-0.56$ PgC yr$^{-1}$ from *in situ* observations and $-0.34$ to $-0.68$ PgC yr$^{-1}$ from satellite retrievals), indicating that the regional inversion system is sufficient to robustly constrain the control vector. Next, the much-refined CMAQ

resolution in JDAS inversion was found to allow for a more detailed characterization of the spatiotemporal distribution of $CO_2$ and to further facilitate an interpretation of carbon flux in a regional context over China. The *a priori* and *a posteriori* seasonal amplitudes ranged from 374.33/333.74, 87.01/80.41, 120.33/113.98, 82.34/88.00 to 413.17/389.48 TgC month$^{-1}$ in north, south, west, central and mainland China, respectively. Also, the drastic fluctuation in the daily variation of *a priori* fluxes

was modified by observational constraints, which appeared more realistic than that of the *a priori* estimates. Moreover, we further investigated the condition of the biosphere carbon sink in several of China's key ecological areas. Using XS as an example, the large transport model errors that were included in the model–data mismatch error involved in previous global inversion studies were effectively reduced by JDAS, and XS was reported to be a relatively stronger sink in contrast to prior



estimates ($-10.12/-10.79$ TgC yr$^{-1}$). Furthermore, the provincial patterns of biosphere flux were investigated and re-estimated. As seen from GOSAT, the difference between the *a posteriori* and *a priori* provincial flux ranged from $-7.03$ TgC yr$^{-1}$ in Heilongjiang to 2.95 TgC yr$^{-1}$ in Shandong. Finally, an evaluation against independent data demonstrated better performance of the *a posteriori* flux when fitted to the non-assimilated $XCO_2$ observations, indicating improved results in the regional

inversion. Considering our prior estimates from CT2019B, the discrepancy could be because our study (a) relied on a fine-scale regional transport model; (b) was constrained by GOSAT $XCO_2$ retrievals with better spatial coverage rather than sparse and inhomogeneous *in situ* observations; (c) performed a joint assimilation of $CO_2$ flux and concentration, which helped reduce the uncertainty in both the initial $CO_2$ fields and the fluxes; and (d) carried out hourly assimilation based on hourly simulation and

observation, which was more realistic.

The regional inversion methodology and results presented here prove the feasibility and superiority of regional CTMs and satellite observations in investigating China's carbon sink. On account of the obvious interannual variation in the biosphere sink, this work also serves as a foundation for future

multi-year retrospective analyses of biosphere–atmosphere exchanges under different meteorological conditions. On the one hand, although the ACOS retrieval technology has been substantially improved and provides unprecedented spatial coverage, more $XCO_2$ retrievals with better quality and lower retrieval uncertainty are still needed, especially during summertime and over west China. On the other hand, a knowledge gap also exists in inversion-based estimates, in which fossil-fuel emissions are

generally assumed to be accurate. Besides uncertainties in natural flux, our current knowledge of urban emissions is far from adequate. Around 70% of fossil-fuel emissions are derived from cities in combination with considerable uncertainties. Within the framework of the Paris Agreement, inversions at higher spatial resolution are an increasing demand, making it crucial to develop the capacity for inversions to quantify urban emissions and assess the effectiveness of emission mitigation strategies,

alongside calls for improvements in observations, *a priori* information, anthropogenic emission inventories, transport models, and inversion technology.





**Acknowledgements**

This work was supported by the National Key Scientific and Technological Infrastructure project
"Earth System Science Numerical Simulator Facility" (EarthLab). This work was also sponsored by the
National Natural Science Foundation of China through Grants 41875014 and 42275153.

**Data Availability**

The GOSAT retrievals were produced by the ACOS/OCO-2 project at the Jet Propulsion Laboratory,
California Institute of Technology, and obtained from the JPL website, co2.jpl.nasa.gov. The
CarbonTracker CT2019B provided by NOAA ESRL, Boulder, Colorado, USA is available from
http://carbontracker.noaa.gov. Data analysis is done with the Matlab version 2019b (MATLAB and
Statistics Toolbox Release, 2019b, mathworks.com) and the Gridded Analysis and Display System
(GrADS; http://cola.gmu.edu/grads/) [Software].

**Competing interests**

The contact author has declared that neither they nor their co-authors have any competing interests.

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

**Figures and Tables**

**Captions:**

**Table 1.**    China's annual carbon sink estimated by different methods, including the inventory method,
ecosystem process models, and atmospheric inversion (unit: PgC yr$^{-1}$). Italic font and gray shading
denote the inversion results after correcting for lateral fluxes according to the flux gap between
top-down and bottom-up estimation. The abbreviations used in the table are as follows: CAMS,



Copernicus Atmosphere Monitoring Service; BI, Bayesian Inversion; JCS, Jena CarboScope; CCDAS,

Carbon Cycle Data Assimilation System; FAPAR, remotely sensed Fraction of Absorbed

Photosynthetically Active Radiation; LMDZ, Laboratoire de Météorologie Dynamique Zoom, a global

transport model; and TM5, the global atmospheric Tracer Model 5.

**Table 2.** Evaluation results between the observations and model (unit: ppm). "$XCO_2$ (validation)"

denotes the independent GOSAT $XCO_2$ retrievals for validation, including model results from CTRL

(black, *a priori* flux simulation), FC (blue, *a posteriori* flux simulation), and AN (red, analysis fields

from JDAS). "$XCO_2$ (assimilation)" represents the observations used for assimilation, and the

corresponding model results come from BG (JDAS background fields). RMSE refers to the

root-mean-square error; CORR refers to the correlation coefficient; MAE refers to the mean absolute

bias; and NUM refers to the $XCO_2$ data amount. The monthly and annual averages were calculated

from the hourly outputs.

**Table 3**.    Probability distribution of hourly bias (unit: %) and bias standard deviation (unit: ppm) of

$XCO_2$ validation including CTRL, FC and AN in 2016.

**Figure 1**. Observation increments ($XCO_2$; unit: ppm) and analysis increments (biosphere flux; unit:

μmole m$^{-2}$ s$^{-1}$) in (a, b) January, (c, d) July, and (e, f) the whole year of 2016.

**Figure 2**. Horizontal distribution of $CO_2$ biosphere fluxes (unit: μmole m$^{-2}$ s$^{-1}$): (a) $E^a$ in 2016, the *a

posteriori* fluxes; (b) $E^a - E^p$ in 2016, the differences between the *a posteriori* and *a priori* $CO_2$ fluxes;

(c) $E^a - E^p$ in January; (d) $E^a - E^p$ in April; (e) $E^a - E^p$ in July; (f) $E^a - E^p$ in October. The red

frames mark west China (28°–48°N, 85°–104°E), north China (37°–52°N, 105°–135°E), central China

(30°–36°N, 105°–120°E), and south China (18°–29°N, 105°–123°E). The blue frames mark six key

ecological areas of China: Daxing'anling (50°–53°N, 121°–127°E); the Loess Plateau (35°–40°N,

105°–112°E); the Qinling Mountains (32°–34°N, 104°–115°E); the rocky desert in Guangxi (22°–25°N,

106°–111°E); Mount Wuyi (26.5°–28.0°N, 117.5°–119.0°E); and Xishuangbanna (21.0°–22.6°N,

100.0°–102.0°E).

**Figure 3**. Time series of $CO_2$ biosphere fluxes over (a) mainland China, (b) west China, (c) north China,

(d) central China, and (e) south China, marked by the red frames in Fig. 2a (unit: TgC month$^{-1}$), in each

month of 2016, obtained from *a priori* values (PR, black), *a posteriori* values (AN, red), and the flux

forecast model (FC, blue). The bars on the right-hand side represent the 12-month average (unit: TgC

month$^{-1}$). The boxes on the left-hand side denote the daily flux (unit: TgC day$^{-1}$), with the whiskers

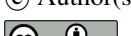



indicating the minimum and maximum and the horizontal lines across the box indicating the 25[th]

percentile, the median, and the 75th percentile, respectively.

**Figure 4**. Time series of $CO_2$ biosphere fluxes over six ecological areas of China (blue frames in Fig.

2a; unit: TgC month$^{-1}$), in each month of 2016, obtained from *a priori* values (PR, black bars) and *a*

*posteriori* values (AN, red bars). The bars on the right-hand side represent the 12-month average (unit:

TgC month$^{-1}$). The subfigures at the bottom denote the daily temperature (blue lines; unit: °C; left-hand

*y*-axis), total solar radiation (red stars; unit: MJ d$^{-1}$; left-hand *y*-axis), and precipitation (grey bars; unit:

mm d$^{-1}$; right-hand *y*-axis), with the right-hand bars representing the annual average.

**Figure 5**. Horizontal distribution of $CO_2$ biosphere fluxes averaged over each province of mainland

China in 2016 (unit: µmole m$^{-2}$ s$^{-1}$): (a) $E^a$: the *a posteriori* fluxes; (b) $E^p$: the *a priori* fluxes; (c)

$E^a - E^p$: the differences between the *a posteriori* and *a priori* $CO_2$ fluxes. Note that Taiwan, Hong

Kong, Macao and Shanghai are not discussed owing to the insufficient grid resolution.

**Figure 6**. The total *a priori* (black) and *a posteriori* (red) $CO_2$ biosphere fluxes over each province of

mainland China in 2016 (unit: TgC yr$^{-1}$). The abbreviations of the provinces are: NM, Neimenggu; SC,

Sichuan; GZ, Guizhou; XJ, Xinjiang; QH, Qinghai; SX', Shaanxi; GX, Guangxi; HL, Heilongjiang; GS,

Gansu; SX, Shanxi; HUN, Hunan; HUB, Hubei; HEB, Hebei; NEN, Henan; JL, Jilin; XZ, Xizang; GD,

Guangdong; JX, Jiangxi; CQ, Chongqing; YN, Yunnan; AH, Anhui; ZJ, Zhejiang; NX, Ningxia; BJ,

Beijing; JS, Jiangsu; SH, Shanghai; FJ, Fujian; TJ, Tianjin; HAN, Hainan; LN, Liaoning; and SD,

Shandong.

**Figure 7**. The annual-averaged horizontal distribution of $CO_2$ concentrations (unit: ppm) near the

surface in 2016: (a) AN: the analysis concentration; (b) FC−AN: the difference between the *a*

*posteriori* flux simulation and analysis concentration fields; (c) CTRL−AN: the difference between the

*a priori* flux simulation and analysis concentration fields; (d) the $XCO_2$ error reduction [see text for

calculation; blue, with the standard deviation (±) of the analysis $XCO_2$ provided] and independent

$XCO_2$ data amount (black stars, rescaled to 1:10) over 8 °–57 °N and 105 °–120 °E at different latitudes.

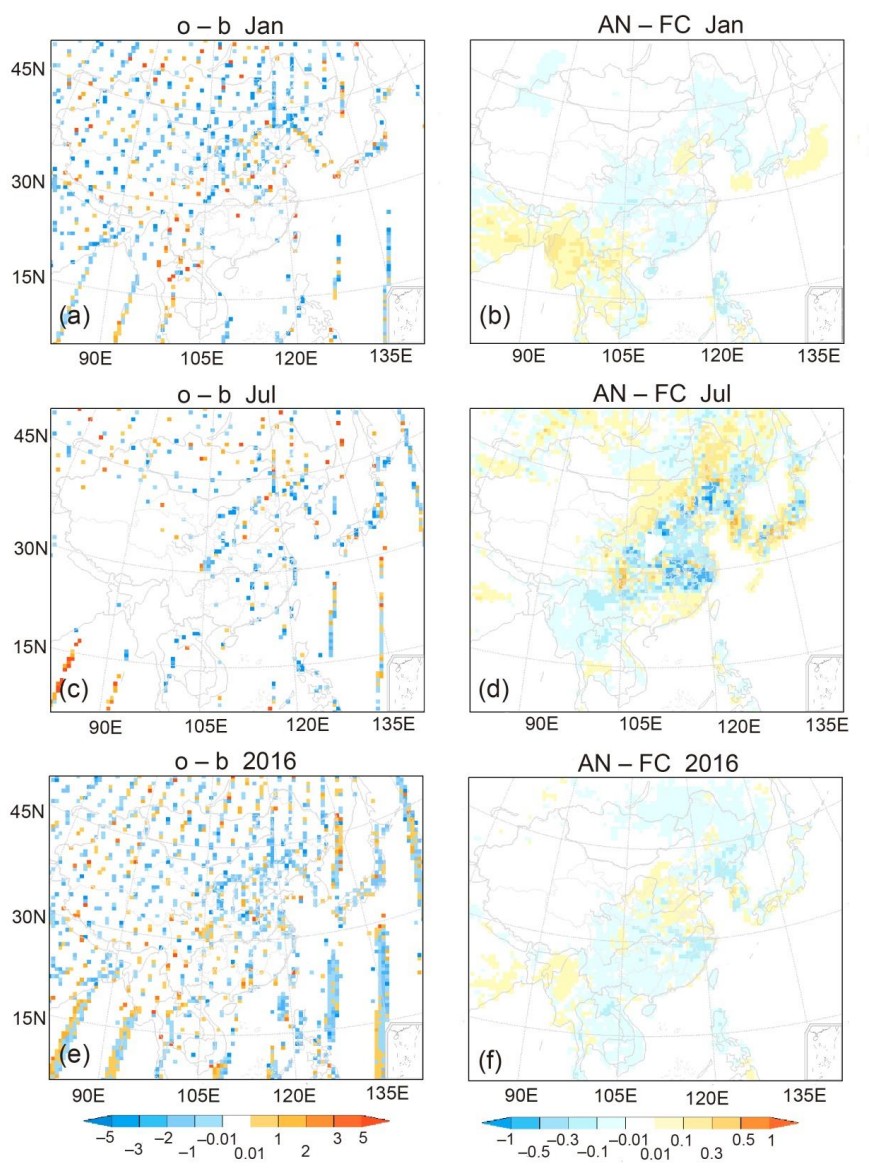

**Figure 1.** Observation increments (XCO$_2$; unit: ppm) and analysis increments (biosphere flux; unit: μmole m$^{-2}$ s$^{-1}$) in (a, b) January, (c, d) July, and (e, f) the whole year of 2016.


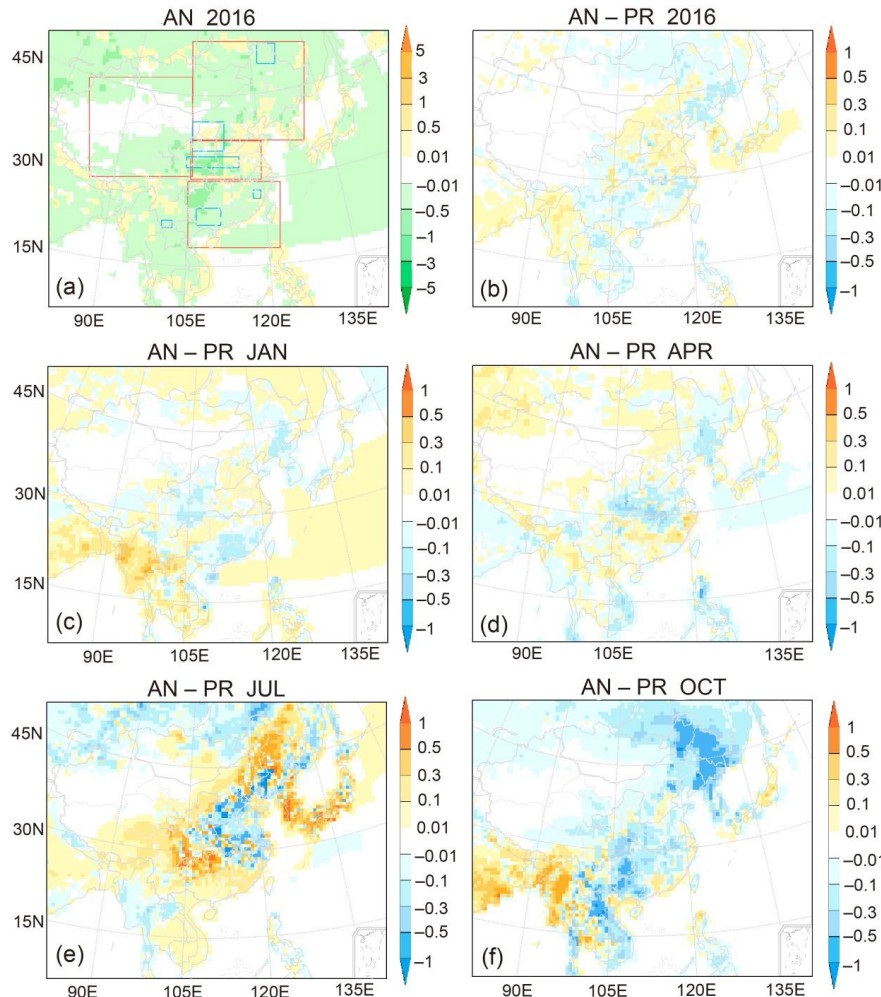

**Figure 2**. Horizontal distribution of $CO_2$ biosphere fluxes (unit: μmole m$^{-2}$ s$^{-1}$): (a) $E^a$ in 2016, the *a posteriori* fluxes; (b) $E^a - E^p$ in 2016, the differences between the *a posteriori* and *a priori* $CO_2$ fluxes; (c) $E^a - E^p$ in January; (d) $E^a - E^p$ in April; (e) $E^a - E^p$ in July; (f) $E^a - E^p$ in October. The red frames mark west China (28°–48°N, 85°–104°E), north China (37°–52°N, 105°–135°E), central China (30°–36°N, 105°–120°E), and south China (18°–29°N, 105°–123°E). The blue frames mark six key ecological areas of China: Daxing'anling (50°–53°N, 121°–127°E); the Loess Plateau (35°–40°N, 105°–112°E); the Qinling Mountains (32°–34°N, 104°–115°E); the rocky desert in Guangxi (22°–25°N, 106°–111°E); Mount Wuyi (26.5°–28.0°N, 117.5°–119.0°E); and Xishuangbanna (21.0°–22.6°N, 100.0°–102.0°E).

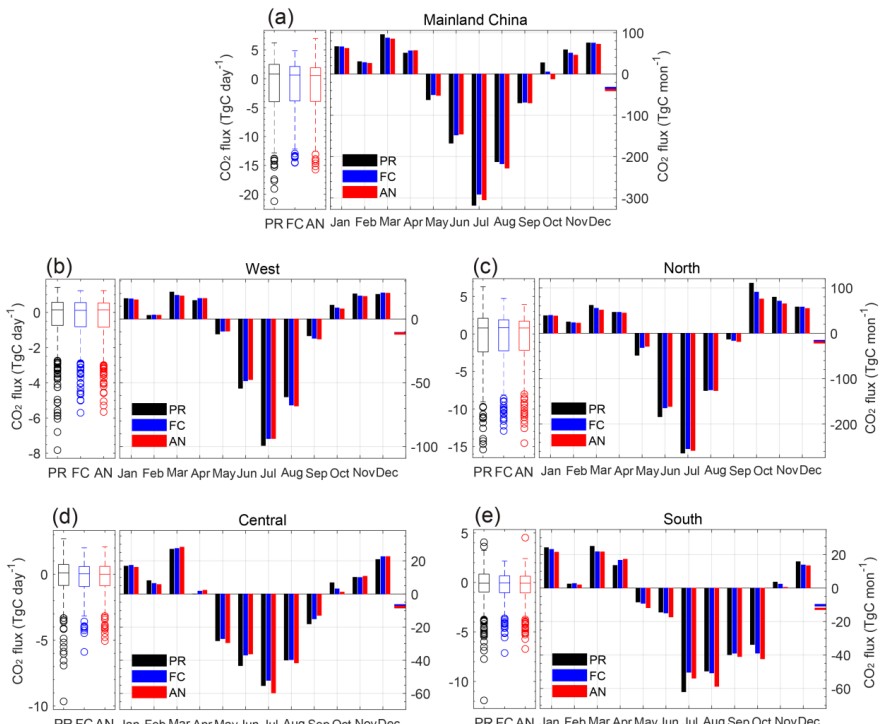

**Figure 3**. Time series of $CO_2$ biosphere fluxes over (a) mainland China, (b) west China, (c) north China, (d) central China, and (e) south China, marked by the red frames in Fig. 2a (unit: TgC month$^{-1}$), in each month of 2016, obtained from *a priori* values (PR, black), *a posteriori* values (AN, red), and the flux forecast model (FC, blue). The bars on the right-hand side represent the 12-month average (unit: TgC month$^{-1}$). The boxes on the left-hand side denote the daily flux (unit: TgC day$^{-1}$), with the whiskers indicating the minimum and maximum and the horizontal lines across the box indicating the 25$^{th}$ percentile, the median, and the 75th percentile, respectively.


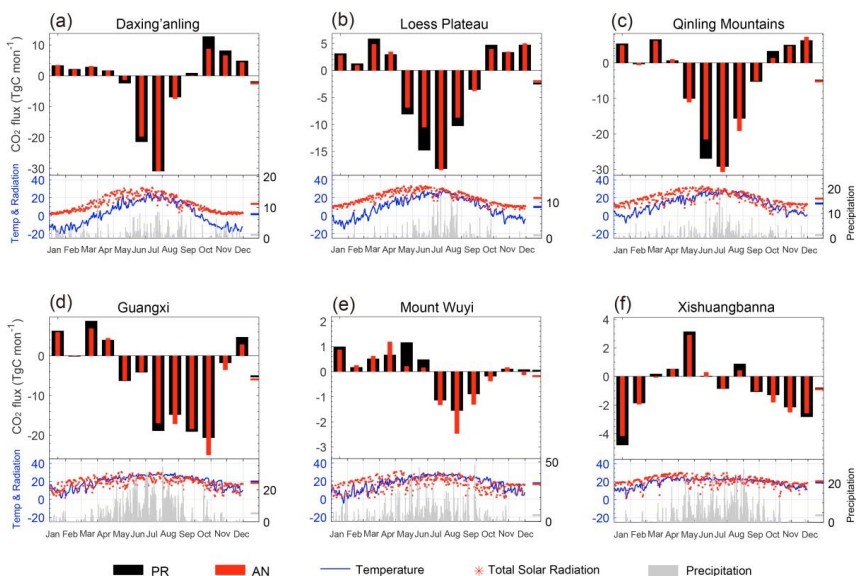


**Figure 4**. Time series of $CO_2$ biosphere fluxes over six ecological areas of China (blue frames in Fig. 2a; unit: TgC month$^{-1}$), in each month of 2016, obtained from *a priori* values (PR, black bars) and *a posteriori* values (AN, red bars). The bars on the right-hand side represent the 12-month average (unit: TgC month$^{-1}$). The subfigures at the bottom denote the daily temperature (blue lines; unit: °C; left-hand *y*-axis), total solar radiation (red stars; unit: MJ d$^{-1}$; left-hand *y*-axis), and precipitation (grey bars; unit: mm d$^{-1}$; right-hand *y*-axis), with the right-hand bars representing the annual average.


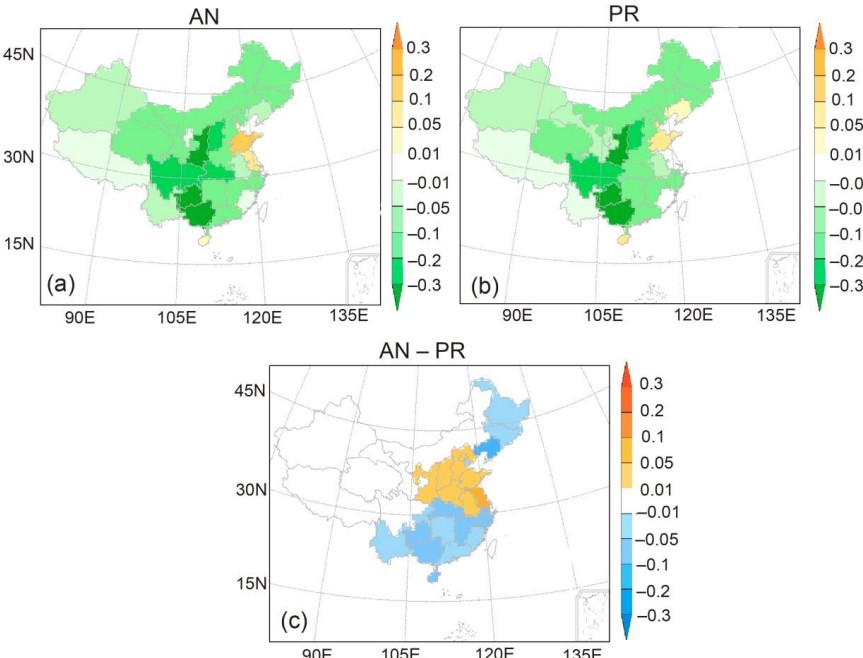

**Figure 5**. Horizontal distribution of $CO_2$ biosphere fluxes averaged over each province of mainland
China in 2016 (unit: μmole $m^{-2}$ $s^{-1}$): (a) $E^a$: the *a posteriori* fluxes; (b) $E^p$: the *a priori* fluxes; (c)
$E^a$–$E^p$: the differences between the *a posteriori* and *a priori* $CO_2$ fluxes. Note that Taiwan, Hong
Kong, Macao and Shanghai are not discussed owing to the insufficient grid resolution.


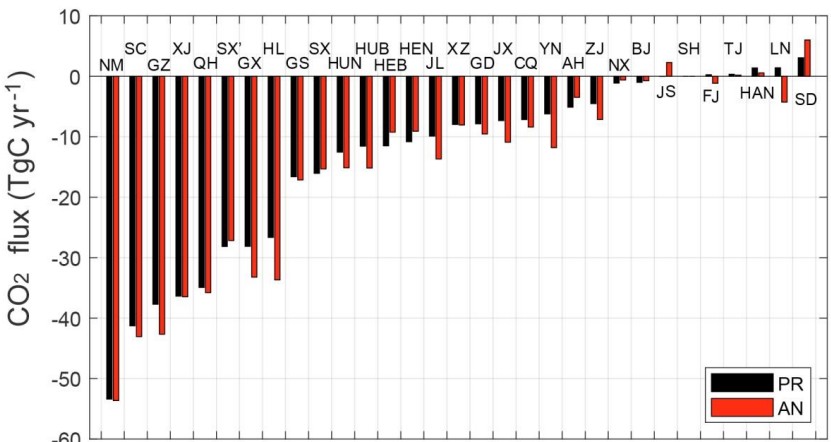

**Figure 6**. The total *a priori* (black) and *a posteriori* (red) $CO_2$ biosphere fluxes over each province of
mainland China in 2016 (unit: TgC $yr^{-1}$). The abbreviations of the provinces are: NM, Neimenggu; SC,
Sichuan; GZ, Guizhou; XJ, Xinjiang; QH, Qinghai; SX', Shaanxi; GX, Guangxi; HL, Heilongjiang; GS,
Gansu; SX, Shanxi; HUN, Hunan; HUB, Hubei; HEB, Hebei; NEN, Henan; JL, Jilin; XZ, Xizang; GD,
Guangdong; JX, Jiangxi; CQ, Chongqing; YN, Yunnan; AH, Anhui; ZJ, Zhejiang; NX, Ningxia; BJ,
Beijing; JS, Jiangsu; SH, Shanghai; FJ, Fujian; TJ, Tianjin; HAN, Hainan; LN, Liaoning; and SD,
Shandong.



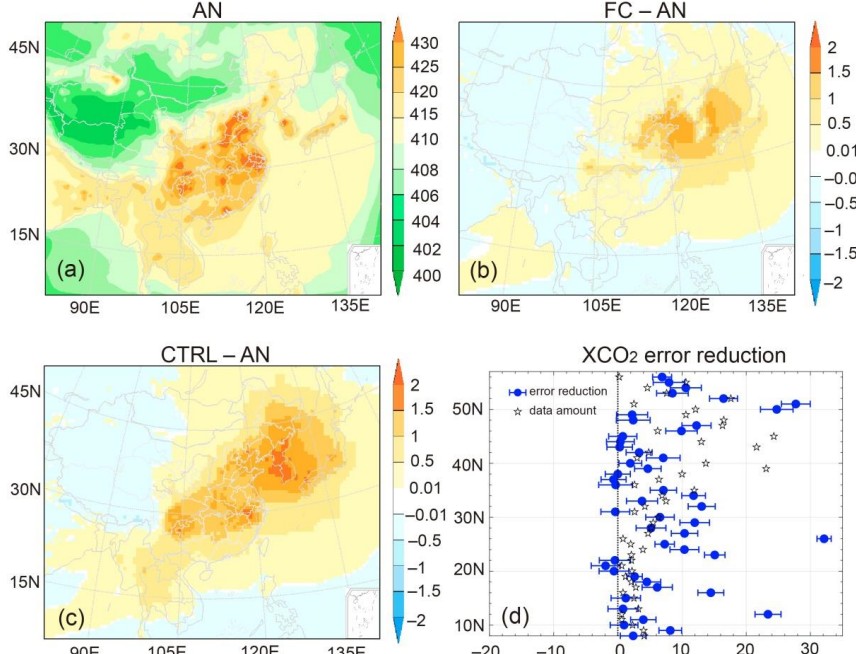

**Figure 7**. The annual-averaged horizontal distribution of $CO_2$ concentrations (unit: ppm) near the surface in 2016: (a) AN: the analysis concentration; (b) FC−AN: the difference between the *a posteriori* flux simulation and analysis concentration fields; (c) CTRL−AN: the difference between the *a priori* flux simulation and analysis concentration fields; (d) the $XCO_2$ error reduction [see text for calculation; blue, with the standard deviation (±) of the analysis $XCO_2$ provided] and independent $XCO_2$ data amount (black stars, rescaled to 1:10) over 8°–57°N and 105°–120°E at different latitudes.



**Table 1.** China's annual carbon sink estimated by different methods, including the inventory method, ecosystem process models, and atmospheric inversion (unit: PgC yr$^{-1}$). Italic font and gray shading denote the inversion results after correcting for lateral fluxes according to the flux gap between top-down and bottom-up estimation. The abbreviations used in the table are as follows: CAMS, Copernicus Atmosphere Monitoring Service; BI, Bayesian Inversion; JCS, Jena CarboScope; CCDAS, Carbon Cycle Data Assimilation System; FAPAR, remotely sensed Fraction of Absorbed Photosynthetically Active Radiation; LMDZ, Laboratoire de Météorologie Dynamique Zoom, a global transport model; and TM5, the global atmospheric Tracer Model 5.


| Method | Carbon sink | Period covered | | | | | Reference |
|---|---|---|---|---|---|---|---|
| Inventory | $-0.18 \pm 0.07$ | 1980–1999 | | | | | Piao et al., 2009 |
| | $-0.29 \pm 0.12$ | 2000–2009 | | | | | Jiang et al., 2016 |
| | $-0.28$ | 2009–2018 | | | | | Wang et al., 2022 |
| Ecosystem process models | $-0.17 \pm 0.04$ | 1980–2002 | | | | | Piao et al., 2009 |
| | $-0.18$ | 1961–2005 | | | | | Tian et al., 2011 |
| | $-0.12 \pm 0.08$ | 1982–2010 | | | | | He et al., 2019 |
| Inversion | | | Observations | Transport models | Optimization | Resolution | |
| *CAMS* | $-0.35 \pm 0.033$ | *1996–2005* | *in situ* $CO_2$ | *LMDZ* | *Bayesian* | *3.75 °×2.5 °, monthly* | *Piao et al, 2009* |
| *CAMS-v19* | $-0.25$ | *2010–2016* | *in situ* $CO_2$ | *LMDZ* | *Variational* | *3.75 °×1.875 °, 8 days,* | *Wang et al., 2022* |
| *BI* | $-0.51 \pm 0.18$ | *2006–2009* | *in situ* $CO_2$ | *TM5* | *Bayesian* | *3 °×2 °, weekly* | *Jiang et al., 2016* |
| *CT-China* | $-0.39 \pm 0.33$ | *2006–2009* | *in situ* $CO_2$ | *TM5* | *EnSRF* | *1 °×1 °, weekly* | *Jiang et al., 2016* |
| CT-China | $-0.33$ | 2001–2010 | in situ $CO_2$ | TM5 | EnSRF | 1 °×1 °, weekly | Zhang et al., 2014 |
| CT-China | $-0.27 \pm 0.20$ | 2010 | in situ $CO_2$ | TM5 | EnSRF | 1 °×1 °, weekly | Chen et al., 2021 |
| CT-China | $-0.41 \pm 0.22$ | 2010–2012 | GOSAT $XCO_2$ | TM5 | EnSRF | 1 °×1 °, weekly | Chen et al., 2021 |
| CT-Europe | $-0.32$ | 2010-2015 | *in situ* $CO_2$ | TM5 | EnSRF | 1 °×1 °, weekly | van der Laan-Luijkx et al., 2017 |
| UoE | $-1.11 \pm 0.38$ | 2010–2016 | *in situ* $CO_2$ | GEOS-Chem | EnKF | 4 °×5 °, 8 days | Wang et al., 2020 |
| UoE | $-0.83 \pm 0.47$ | 2010–2015 | GOSAT $XCO_2$ | GEOS-Chem | EnKF | 4 °×5 °, 8 days | Wang et al., 2020 |
| UoE | $-0.68$ | 2015 | OCO-2 $XCO_2$ | GEOS-Chem | EnKF | 2 °×2.5 °, 8 days | Schuh et al., 2022 |
| JCS | $-0.48$ | 2010-2015 | *in situ* $CO_2$ | TM3 | Bayesian | 4 °×5 °, monthly | Rödenbeck et al., 2018 |
| GCASv2 | $-0.34 \pm 0.14$ | 2010–2015 | GOSAT $XCO_2$ | MOZART-4 | EnSRF | 1 °×1 °, weekly | He et al., 2022 |
| CCDAS | $-0.43 \pm 0.09$ | 2010–2015 | *in situ* $CO_2$, FAPAR | TM2 | 4D-Var | 2 °×2 °, monthly | He et al., 2022 |
| CT-2019B | $-0.43$ | 2016 | *in situ* $CO_2$ | TM5 | EnSRF | 1 °×1 °, weekly | Jacobson et al., 2020 |
| JDAS | $-0.68$ | 2016 | *in situ* $CO_2$ | CMAQ | EnSRF | 64×64km, hourly | This study |
| JDAS | $-0.47$ | 2016 | GOSAT $XCO_2$ | CMAQ | EnSRF | 64×64km, hourly | This study |






**Table 2.** Evaluation results between the observations and model (unit: ppm). "XCO$_2$ (validation)" denotes the independent GOSAT XCO$_2$ retrievals for validation, including model results from CTRL (black, *a priori* flux simulation), FC (*italic*, *a posteriori* flux simulation), and AN (**bold**, analysis fields from JDAS). "XCO$_2$ (assimilation)" represents the observations used for assimilation, and the corresponding model results come from BG (JDAS background fields). RMSE refers to the root-mean-square error; CORR refers to the correlation coefficient; MAE refers to the mean absolute bias; and NUM refers to the XCO$_2$ data amount. The monthly and annual averages were calculated from the hourly outputs.

| | XCO$_2$ (validation) | | | | XCO$_2$ (assimilation) | | | | |
| | RMSE | CORR | MAE | NUM | NUM | RMSE | CORR | MAE | Median of |
| | (CTRL/*FC*/**AN**) | (CTRL/*FC*/**AN**) | (CTRL/*FC*/**AN**) | | | (BG) | (BG) | (BG) | XCO$_2$ uncertainty |
|---|---|---|---|---|---|---|---|---|---|
| Jan | 3.80/*3.79*/**2.45** | 0.19/*0.19*/**0.46** | 2.45/*2.45*/**2.05** | 2024 | 1788 | 2.38 | 0.53 | 1.97 | 0.66 |
| Feb | 2.42/*2.40*/**2.37** | 0.42/*0.42*/**0.43** | 1.99/*1.98*/**1.97** | 1902 | 1870 | 2.29 | 0.52 | 1.87 | 0.72 |
| Mar | 2.48/*2.46*/**2.40** | 0.36/*0.37*/**0.38** | 2.05/*2.03*/**2.00** | 1409 | 1617 | 2.26 | 0.49 | 1.83 | 0.78 |
| Apr | 1.90/*1.90*/**1.79** | 0.31/*0.32*/**0.35** | 1.91/*1.91*/**1.84** | 1037 | 1346 | 2.18 | 0.36 | 1.76 | 0.91 |
| May | 2.70/*2.71*/**2.47** | 0.18/*0.18*/**0.17** | 2.23/*2.23*/**2.10** | 826 | 1090 | 2.36 | 0.16 | 1.95 | 0.91 |
| Jun | 2.34/*2.35*/**2.26** | 0.70/*0.70*/**0.73** | 1.84/*1.83*/**1.82** | 615 | 734 | 2.21 | 0.72 | 1.78 | 0.97 |
| Jul | 2.45/*2.44*/**2.37** | 0.82/*0.82*/**0.83** | 2.02/*2.02*/**1.98** | 560 | 728 | 2.41 | 0.80 | 1.99 | 0.99 |
| Aug | 2.49/*2.50*/**2.42** | 0.65/*0.65*/**0.66** | 2.03/*2.03*/**2.01** | 742 | 842 | 2.38 | 0.69 | 1.98 | 0.95 |
| Sep | 2.26/*2.22*/**2.11** | 0.37/*0.38*/**0.43** | 1.82/*1.80*/**1.71** | 879 | 854 | 2.15 | 0.47 | 1.76 | 0.82 |
| Oct | 2.37/*2.28*/**2.22** | 0.37/*0.40*/**0.44** | 1.91/*1.86*/**1.84** | 1192 | 1190 | 2.29 | 0.45 | 1.88 | 0.75 |
| Nov | 2.39/*2.36*/**2.25** | 0.54/*0.55*/**0.58** | 1.91/*1.89*/**1.84** | 1627 | 1517 | 2.27 | 0.60 | 1.85 | 0.67 |
| Dec | 2.36/*2.35*/**2.34** | 0.52/*0.52*/**0.53** | 1.94/*1.93*/**1.91** | 1847 | 1688 | 2.26 | 0.60 | 1.85 | 0.64 |
| 2016 | 2.65/*2.63*/**2.34** | 0.66/*0.66*/**0.73** | 2.03/*2.02*/**1.93** | 14660 | 15264 | 2.29 | 0.72 | 1.87 | 0.77 |




**Table 3**. Probability distribution of hourly bias (unit: %) and bias standard deviation (unit: ppm) of XCO$_2$ validation including CTRL, FC and AN in 2016.

| Bias probability distribution | CTRL | FC | AN |
|---|---|---|---|
| [-4,4] | 89.64 | 89.89 | 91.02 |
| [-3,3] | 75.63 | 75.99 | 76.84 |
| [-2,2] | 56.13 | 56.45 | 56.88 |
| [-1,1] | 30.22 | 30.08 | 30.24 |
| [0,4] | 53.43 | 53.62 | 55.74 |
| [0,3] | 44.65 | 44.86 | 46.21 |
| [0,2] | 32.26 | 32.46 | 33.07 |
| Bias standard deviation | 2.6268 | 2.6072 | 2.2674 |