# Peer review of "The carbon sink in China as seen from GOSAT with a regional inversion system based on CMAQ and EnKS"

_Atmospheric Chemistry and Physics, 2022_

## Author Comment (AC1)

**Response to Reviewer #1**

We thank the reviewer#1 for the insightful and detailed comments and suggestions, which helped to significantly improve the manuscript. The reviewer's comments are shown in *blue italics* with the author responses in black.

*General comment:*

*Kou et al. estimated biosphere carbon fluxes over China by applying a regional inversion system to GOSAT $CO_2$ data. The inversion was designed to provide a higher spatialtemporal resolution than previous studies. While the topic is definitely interesting to the reader of ACP, the manuscript, in its current form, is not up to the standard. My main comments are as follows.*

*(1) The paper lacks technical rigor. The authors present the high spatial-temporal resolution (64 km and 1 hour) as the innovation of the paper, but do not provide justification that the inversion of GOSAT $CO_2$ data can meaningfully resolve hourly data, as GOSAT observations are daily observations at the same local solar time. A reader may be interested in quantitative information on to what extent the results are affected by prior information and to what extent they are constrained by observations. In addition, the authors claimed that the inversion is verified against "independent observations". But in fact these validation data are also taken from the same GOSAT $CO_2$ dataset. Although these observations are not assimilated in the inversion, they may well have error distributions similar to those assimilated. Hence, these data cannot be regarded as "independent" validation data.*

Thank the review for the comment. First of all, measured $CO_2$ concentrations are the result of upstream surface fluxes and atmospheric transport process. Generally speaking, the longer in the past a flux event occurred, the smaller its impact will be on a given sample of air. Therefore, we choose an "assimilation window" to represent how far back in time we expect to be able to pinpoint a given flux signal from available measurements. In an assimilation cycle, the fluxes for the 24-h assimilation window have been designed to be optimized hour by hour successively in this study. Accordingly, the fluxes have been adjusted 24 times before generating posterior fluxes. Actually, in this study the NOAA operational EnKF system, which is an EnSRF and modified with the ensemble Kalman smoother (EnKS) feature, is further extended to jointly assimilate the $CO_2$ concentrations and fluxes to update

the flux and concentration fields, respectively. The EnKS allows for a sequential processing of the measurements in time, which updates the ensemble at prior times every time new observations are available. Thus, EnKS that can take into future observations into account is used to assimilate the concentrations and update the fluxes.

In this study, the state vector $\mathbf{x}$ includes the mass concentration $\mathbf{C}$ and the flux $\boldsymbol{E}$, i.e. $\mathbf{x} = [\boldsymbol{C}, \boldsymbol{E}]^{\mathrm{T}}$. Here, the state variables of mass concentration $\mathbf{C}$ are the $CO_2$ concentrations. The ensemble forecast concentration fields of $CO_2$ are respectively used in calculating ensemble fluxes $\boldsymbol{E}_{i,t}^{f}$ as described in Section 2.2.1. The ensemble members of $CO_2$ concentration fields $\mathbf{C}^{\mathrm{f}}$ are forecasted using CMAQ, forced by the forecast emissions $\mathbf{E}^{\mathrm{f}}$ whose initial conditions are previously analyzed concentration fields. Now, the background of the joint vector, $\mathbf{x}^{\mathrm{f}} = \left[\mathbf{C}^{\mathrm{f}}, \mathbf{E}^{\mathrm{f}}\right]^{\mathrm{T}}$, has been produced. Then, the analyzed state vector, $\mathbf{x}^{\mathrm{a}} = [\mathbf{C}^{\mathrm{a}}, \boldsymbol{E}^{\mathrm{a}}]^{\mathrm{T}}$, is optimized by applying EnKS. The configurations of the EnKS were as follows: 1) ensemble size was set to 50; 2) the horizontal localization radius was 1280 km; 3) the covariance inflation factor β was set to 80; 4) the assimilation window in EnKS was set to 24 h (Peng et al., 2023). In addition, hour-by-hour assimilation was adopted attribute to the novel flux forecast model, fine-scale CMAQ forward hourly simulation output, as well as the available GOSAT observations at certain hour of the day. Therefore, in spite of the daily GOSAT observations at the same local solar time, the inversion of GOSAT $CO_2$ data can meaningfully resolve hourly data through the EnKS configuration.

Furthermore, readers may be interested in quantitative information on to what extent the results are affected by prior information and to what extent they are constrained by observations. Usually, it is hard to evaluate the optimized flux, because comparison with *in situ* flux measurements is difficult on account of the discrepancy in scales between assimilated fluxes in the model grid and eddy-covariance measurements over a very large uniform underlying surface. Nevertheless, the prior information has been embodied in *a priori* flux simulated concentrations, and observation information has been embodied in the *a posteriori* flux simulation, whose fluxes are constrained by observations. By evaluating the differences between these two sets of simulation results, the prior information and observation information now have access to be accessed quantitatively. Therefore, this traditional

approach was adopted as a compromise to assess whether the *a posteriori* fluxes would enable improvements in the fit to the observed $CO_2$ concentrations. The RMSEs of prior and posterior simulations (i.e. CTRL and FC) are further presented in Table R1 in the revised manuscript. According to the quantitative evaluation on RMSE, the observations have played a positive role in improving carbon sink over the model domain. At the site scale, some sites tend to systematically be poorly fitted by the inversions, in particular those in the vicinity of large urban areas with large anthropogenic emissions, such as Jinsha and Lin'an. Besides these two sites, the difference between CTRL and FC is affected by the observation information through assimilation ranges from 0.25% to 12.34% (i.e. RMSE decreasing rates), with an average of 2.48% among all surface observation sites. Moreover, smaller correlation coefficient improvement in the contrast of CTRL and FC imply that prior flux patterns play an important role in the $CO_2$ variation compared with that of posterior flux (Table 1).

In addition, although some GOSAT observations which are not assimilated in the inversion were used as independent data to evaluate the posterior flux, they may have error distributions similar to those assimilated. Therefore, surface *in situ* observations from 14 sites are further used as independent observations to evaluate the inversion result in the revised manuscript. Comparison between surface observations, prior flux simulation, posterior flux simulation and the analysis for hourly $CO_2$ concentration is added in Table R1. The modeled $CO_2$ concentrations were extracted from the simulated hourly $CO_2$ fields according to the locations, elevation, and time of each observation. The averages of observation, CTRL, FC, and AN over these 14 stations are 410.97, 413.01, 412.82, and 412.21 ppm, respectively. According to the statistics listed in Table R1, the statistics of the analytical field (AN) are better than FC and CTRL, including RMSE and MAE, which gives a direct indication that the assimilation performs well. Taking improvement as example, the RMSE improvement rate between the FC and CTRL mostly ranges from –2.13% to 12.34% with an average of 2.48%, and the MAE improvement rate ranges from 0.08% to 9.73% with an average of 2.37%. Further, although the RMSE and MAE of AN are lower than CTRL, those of FC are higher than CTRL in Lin'an and Jinsha. This could be attributed to the influence from human activities to a large extent (Liang et al., 2023), because Jinsha and Lin'an are both urban background stations for Central China (i.e. Jinsha locates in Wuhan, Hubei) and East China (i.e. Lin'an locates in Yangtze River Delta). Thus, this helps to check that the inversions actually improve the model fits to the observations but also to determine whether

Lin'an and Jinsha sites are particularly problematic for natural flux inversions.

We have modified the relevant parts in the revised manuscripts (Line 189−270, Line 595−615, and Line 500−535), and Table 4 is further added and discussed.

**Table R1.** Evaluation results between *in situ* observations and model, including CTRL (black, *a priori* flux simulation), FC (*italic*, *a posteriori* flux simulation), and AN (**bold**, analysis fields from JDAS).

| | Lat.(°N) /Lon.(°E) | OBS. NUM | OBS. Freq. | RMSE (CTRL/FC/AN) | RMSE Imp. Rate FC/AN (%) | MAE (CTRL/FC/AN) | MAE Imp. Rate FC/AN (%) | General Site Description |
|---|---|---|---|---|---|---|---|---|
| Longfengshan | 44.73/127.60 | 840 | Hourly | 10.94/*10.87*/**10.38** | *0.63*/**5.16** | 7.83/*7.81*/**7.72** | *0.30*/**1.40** | Forest (Northeast China) |
| Shangdianzi | 40.65/117.12 | 1620 | Hourly | 10.00/*9.87*/**9.74** | *1.34*/**2.58** | 6.87/*6.62*/**6.64** | *3.53*/**3.26** | Cropland (North China) |
| Mt. Waliguan | 36.28/100.90 | 338 | Daily | 7.05/*6.64*/**6.31** | *5.78*/**10.43** | 4.63/*4.38*/**4.15** | *5.35*/**10.35** | Tibet Plateau (China) |
| Shangri-La | 28.00/99.40 | 1709 | Hourly | 9.76/*9.62*/**9.44** | *1.42*/**3.21** | 7.21/*7.08*/**7.02** | *1.72*/**2.61** | Forest (Southeast China) |
| Lin'an | 30.30/119.72 | 1410 | Hourly | 9.42/*9.49*/**8.60** | *−0.73*/**8.70** | 6.63/*6.78*/**6.14** | *−2.16*/**7.45** | Forest (East China) |
| Jinsha | 29.63/114.22 | 30 | Weekly | 9.21/*9.41*/**8.94** | *−2.13*/**2.96** | 6.96/*7.04*/**6.46** | *−1.15*/**7.13** | Urban (Central China) |
| King's Park | 22.31/114.17 | 364 | Daily | 22.12/*21.63*/**21.10** | *2.22*/**4.63** | 17.02/*16.68*/**16.06** | *1.98*/**5.06** | Urban (Hong Kong, China) |
| Ulaan Uul | 44.45/111.08 | 49 | Weekly | 5.50/*5.41*/**5.22** | *1.62*/**5.06** | 3.70/*3.63*/**3.52** | *2.02*/**5.09** | Grassland (Mongolia) |
| Ryori | 39.03/141.82 | 8553 | Hourly | 6.85/*6.77*/**6.06** | *1.08*/**11.51** | 4.59/*4.48*/**3.91** | *2.21*/**14.68** | Mountain (Japan) |
| Mt. Dodaira | 36.00/139.20 | 7928 | Hourly | 7.62/*7.51*/**7.12** | *1.45*/**6.50** | 5.37/*5.31*/**5.00** | *1.22*/**6.95** | Mountain (Japan) |
| Kisai | 36.08/139.55 | 8686 | Hourly | 17.09/*15.90*/**15.80** | *6.99*/**7.56** | 13.00/*12.22*/**12.24** | *5.99*/**5.83** | Urban (Japan) |
| Anmyeon-do | 36.53/126.32 | 3228 | Hourly | 16.00/*14.03*/**13.81** | *12.34*/**13.70** | 10.42/*9.41*/**8.85** | *9.73*/**15.06** | Coastal (Korea) |
| Jeju Gosan | 33.30/126.21 | 4373 | Hourly | 10.10/*9.85*/**8.79** | *2.42*/**12.97** | 7.29/*7.12*/**6.34** | *2.39*/**13.10** | Ocean (Korea) |
| Yonagunijima | 24.47/123.02 | 8085 | Hourly | 9.24/*9.21*/**8.60** | *0.25*/**6.86** | 7.39/*7.38*/**6.91** | *0.08*/**6.41** | Ocean (Japan) |
| AVE | | | | 10.78/*10.44*/**9.99** | *2.48*/**7.27** | 7.78/*7.57*/**7.21** | *2.37*/**7.49** | |

Note. 'Lat./Lon.' refers to the latitude and longitude of site; 'OBS. NUM' refers to the observation amount; 'OBS. Freq.' refers to the observation time frequency; 'RMSE Imp. Rate' refers to the improvement rate of RMSE, i.e., $(RMSE_{CTRL}−RMSE_{FC})/RMSE_{CTRL}×100\%$ and $(RMSE_{CTRL}−RMSE_{AN})/RMSE_{CTRL}×100\%$; 'MAE Imp. Rate' refers to the improvement rate of MAE, i.e., $(MAE_{CTRL}−MAE_{FC})/MAE_{CTRL}×100\%$ and $(MAE_{CTRL}−MAE_{AN})/MAE_{CTRL}×100\%$, respectively. The annual averages were calculated from the hourly output.

Here are the above-mentioned references.

Liang, M., Zhang, Y., Ma, Q., L., Yu, D. J., Chen, X. J., & Cohen, J. B. (2023). Dramatic decline of observed atmospheric $CO_2$ and $CH_4$ during the COVID-19 lockdown over the Yangtze River Delta of China. *Journal of Environmental Sciences*, 124, 712−722, https://doi.org/10.1016/j.jes.2021.09.034

Peng, Z., Kou, X. X., Zhang, M. G., Lei, L. L., Miao, S. G., Wang, H. M., Jiang, F., Han, X., and Fang, S. X. (2023). $CO_2$ flux inversion with a regional joint data assimilation system based on CMAQ, EnKS, and surface observations. *Journal of Geophysical Research-Atmosphere*, 128, e2022JD037154. https://doi. org/10.1029/2022JD037154

*(2) The writing needs to be improved. For example, Section 2.2 (a key section describing the inversion algorithm) is difficult to follow. The logic flow is not clear. Important information such as how the error covariance matrices are specified and updated in the ESRFs is missing. Results in Section 3.3-3.4 are not presented in a concise and well-structured way. The discussion is not focused on new findings and insight, but in many cases, reporting numbers without proper interpretation. There are several occurrences where some discussion points and even exact same sentences are repeated. For example, "(the system) is sufficient to robustly constrain the control vector" appears in line 26, 416, and 624. Notation and terminologies are used inconsistently and loosely, for instance, control vectors, state vector, and state variables are all used to represent a similar concept without explicit definitions. Overall, I'd suggest to substantially shorten the paper to focus on the contribution of this study to the field. Attention needs to be paid to logic flows and consistent terminology.*

Thank the reviewer for the comment. First of all, Section 2.2 has been revised to describe the inversion algorithm. In the joint assimilation framework, besides the application of CMAQ to generate ensemble $CO_2$ concentrations, a flux forecast model was also designed to represents flux variations on account of fluxes acting as model forcing. The EnKS was further designed to joint assimilate $CO_2$ concentrations and fluxes. A brief description of the flux forecast model as well as the ensemble assimilation scheme is presented in Section 2.2. Consequently, after completing the "forecast step", Kalman gain matrix $K$ is obtained by minimizing the analysis error covariance with evolved forecast error covariance over time. Then, the associated analyzed state variables, $\boldsymbol{x}^a = \left[ \boldsymbol{C}^a, \boldsymbol{E}^a \right]^T$, can be updated by applying the EnKS constrained by GOSAT retrievals in the "analysis step". In addition, the distribution of ensemble spread of $CO_2$ flux in January 2016 is provided in Figure R1. It shows that the values of the ensemble spread ranges from 0.2 to 0.8 in most areas, which are consistent with our previous studies (Peng et al., 2015 in Figure 11c and Peng et al., 2023).

Furthermore, detailed modifications have been made in Section 3.3–3.4 as well as the full text to present in a concise and well-structured way. And we have separated the results and discussion in the revised manuscript. Repeated points and sentences have been carefully considered and revised. Moreover, the notation and terminologies are redefined clearly and revised to keep consistency. For instance, control vectors, state vectors, joint vectors and state variables are modified as state variables, which refers to the variables used to assimilation (i.e., $x = \begin{bmatrix} C, E \end{bmatrix}^T$ ).

We have modified the relevant parts in the revised manuscripts (Section 2, Section 3 and Section 4).

[Figure]

Figure R1. The ensemble spread of $\lambda_{i,t}^a$ at model level 1 in January 2016, when $\beta$=80.

Here are the above-mentioned references.

Peng, Z., Zhang, M. G., Kou, X. X., Tian, X. J., & Ma, X. G. (2015). A regional carbon flux data assimilation system and its preliminary evaluation in East Asia. *Atmospheric Chemistry and Physics*, 15, 1087–1104. https://doi.org/10.5194/acp-15-1087-2015.

Peng, Z., Kou, X. X., Zhang, M. G., Lei, L. L., Miao, S. G., Wang, H. M., Jiang, F., Han, X., and Fang, S. X. (2023). $CO_2$ flux inversion with a regional joint data assimilation system based on CMAQ, EnKS, and surface observations. *Journal of Geophysical Research-Atmosphere*, 128, e2022JD037154. https://doi. org/10.1029/2022JD037154

*Minor comments:*

*Line 23, 228: What is an observational operator? It is never clearly defined.*

Observation operator converts the background forecast to observation space. To obtain the simulated

observations $H(\boldsymbol{C}^f)$, observation operator performs the necessary interpolation and transformation from model 3D $CO_2$ concentrations forecast to observation space $XCO_2$. The simulated $CO_2$ concentration profiles were mapped into the satellite retrieval levels and then vertically integrated based on the satellite averaging kernel according to the Equation 2. In addition, for the $H(\boldsymbol{E}^f)$, it should be noted that observation operator includes not only interpolation (i.e. Equation 2) but also CMAQ simulation to convert from flux to simulated $XCO_2$.

We have modified the description of observation operator in the revised manuscripts (Line 235−250), and we hope we can make the meaning clear now.

*Line 111-112. The author first claimed that "regional CTMs are rarely used in satellite carbon data assimilation" but then cite a few studies that performed regional carbon data assimilation, which appears to be inconsistent. Moreover, the authors need to clarify what are innovations in this study relative to these cited studies.*

At present, almost all China's carbon sink inversions use global atmospheric transport models with a relatively coarse spatial resolution and long timescale from a weekly or monthly perspective. For instance, Huang and Zhang assimilated $CO_2$ observations with regional CTM to optimize the $CO_2$ concentration fields (Huang et al., 2014; Zhang et al., 2021). In recent years, several studies have used regional CTMs in $CO_2$ flux inversions inferred from surface stations, towers, and aircraft flights. The potential use of regional CTM in $CO_2$ inversions with satellite has been explored with artificial retrievals by observing system simulation experiments. Thus, regional CTMs has been rarely used in real satellite carbon data inversion of China's terrestrial carbon sink, even though multi-model comparisons have reported large uncertainties introduced by global CTMs in estimating the carbon sink. Because of this, taking advantage of regional chemistry transport models for mesoscale simulation and spaceborne sensors for spatial coverage, the GOSAT $XCO_2$ retrievals were introduced in CMAQ and EnKS-based regional inversion system to constrain China's biosphere sink.

We are sorry about the confusion in the introduction. We have modified the abstract in the revised manuscript (Line 110−130), and the motivation and innovation has been rewritten in the introduction.

*Line 140: The study uses historical GOSAT observations not "real-time" GOSAT observations.*

Yes, this study uses historical GOSAT observations. The "real-time" statement has been modified in the revised text and we have checked the full text with the incorrect expression.

*Line 150-154: Two science questions are raised by the end of the Introduction, but it is not apparent that the discussion is focused on these questions nor these questions are adequately addressed.*

Yes, the reviewer makes a good point. Actually, this paper focuses on the following two questions:

1. On what scales can regional CTMs and GOSAT observations facilitate the inversion of China's carbon sink?

2. What is the difference between posterior flux inferred from spaceborne retrievals and prior flux?

For Question 1, we try to discussion the posterior flux from country, regional to provincial scales, as well as daily, monthly, and annual variation. First of all, we found that the size of the assimilated biosphere sink in China was $-0.47$ PgC yr$^{-1}$, which was comparable with previous results (i.e., $-0.27$ to $-0.68$ PgC yr$^{-1}$). Furthermore, the seasonal patterns were recalibrated well, with a growing season that shifted earlier in the year over central and south China. We further investigated the condition of the biosphere carbon sink in several of China's key ecological areas. Using Xishuangbanna (XS) as an example, the large transport model errors that were included in the model–data mismatch error involved in previous global inversion studies were effectively reduced by JDAS, and XS was reported to be a relatively stronger sink in contrast to prior estimates ($-10.79/-10.12$ TgC yr$^{-1}$). Moreover, the provincial-scale biosphere flux was re-estimated, and the difference between the *a posteriori* and *a priori* flux ranged from $-7.03$ TgC yr$^{-1}$ in Heilongjiang to $2.95$ TgC yr$^{-1}$ in Shandong.

For Question 2, considering our prior estimates from CT2019B, the discrepancy between posterior and prior flux could be because our study (a) relied on a fine-scale regional transport model; (b) was constrained by GOSAT XCO$_2$ retrievals with better spatial coverage rather than sparse and inhomogeneous *in situ* observations; (c) performed a joint assimilation of CO$_2$ flux and concentration, which helped reduce the uncertainty in both the initial CO$_2$ fields and the fluxes; and (d) carried out hourly assimilation based on hourly simulation and observation, which was more realistic. In addition,

we further assess the performance of the *a posteriori* $CO_2$ fluxes by comparing the CTRL, FC and AN concentration against independent observations. In addition, we further assess the performance of the *a posteriori* $CO_2$ fluxes by comparing the CTRL, FC and AN results again 14 surface in-situ observations sites. The averages of observation, CTRL, FC, and AN over these 14 stations are 410.97, 413.01, 412.82, and 412.21 ppm, respectively. According to the statistics listed in Table 4, the statistics of the analytical field (AN) are better than FC and CTRL, including RMSE and MAE, which gives a direct indication that the assimilation performs well. Taking improvement as example, the RMSE improvement rate between the FC and CTRL mostly ranges from 0.25% to 12.34% (besides two sites in the vicinity of large urban areas) with an average of 2.48%, and the MAE improvement rate ranges from 0.08% to 9.73% with an average of 2.37%.

We have rephrased the questions, address the questions, and modified the relevant parts in the revised manuscript.

*Line 171: Why does not CMAQ need initial and lateral boundary meteorological fields. Is CMAQ coupled with a meteorology model (e.g., WRF)? A typical regional chemical transport model like CMAQ is driven by archived met fields and does not need initial and lateral boundary meteorological fields.*

Thank the reviewer for the comment. The $CO_2$ concentration was forecast with the regional atmospheric chemistry transport model, CMAQ, coupled with the RAMS for providing the meteorological fields. On one hand, CMAQ, a regional CTM, needs initial and boundary $CO_2$ concentrations fields, which is extracted from CT2019B concentration products ($3\degree \times 2\degree$, 3h, globally). One the other hand, CMAQ is coupled with a regional meteorology model (i. e., RAMS), and CMAQ is driven by archived meteorological fields from RAMS. RAMS provides the three-dimensional meteorological fields in same horizontal resolution with CMAQ, with the lowest seven layers being the same as those in CMAQ. Consequently, RAMS need initial and boundary meteorological fields. In this study, the initial and lateral boundary meteorological fields, sea surface temperatures, and initial soil conditions were prescribed by European Centre for Medium-Range Weather Forecasts reanalysis data with a spatial resolution of $1\degree \times 1\degree$ and 6-hourly temporal intervals.

We have modified the relevant parts of the revised manuscript (Line 165−190); please check if it is clear now.

*Line 174: What is "'real' initial and lateral boundary atmospheric CO2 concentrations"?*

Considering the unique characteristics (e.g., long atmospheric lifetime, large background concentration, and strong biosphere–atmosphere exchanges) of atmospheric $CO_2$ that are distinctly different from other traditionally modeled chemical pollutants, some key requirements for regional $CO_2$ modeling have been noted, such as using realistic initial and lateral boundary conditions (Kou et al., 2015). In this study, the initial fields and boundary conditions of atmospheric $CO_2$ volume fraction were obtained by interpolation of NOAA's CT2019B ($3\,°\times 2\,°$, 3h, globally). CT2019B is a widely recognized product, whose global $CO_2$ concentrations were created using the optimized surface fluxes and simulated atmospheric transport of CarbonTracker. CT2019B could represent the optimum estimate of the global distribution of atmospheric $CO_2$ (Jacobson et al., 2020).

We have modified the relevant parts of the revised manuscript (Line 167−171).

*Line 232: If yf and yp are "wet" CO2 concentration, you should apply (1-w)^-1 to convert "wet" concentration to "dry" concentration, instead of multiplying (1-w)*

Thank the reviewer for the comment. We're sorry about the mistake in typing the formula. Actually, we applied $(1-w)^{-1}$ to convert "wet" concentrations to "dry" concentration, as suggested by Feng et al. (2009). We have modified the formula in the revised manuscript (Line 241).

*Line 245: What is BG here?*

BG denotes the model's first guess background fields $x^f$ in the assimilation scheme. The background of the state variables, $x^f = \left[C^f, E^f\right]^T$, can be prepared by CMAQ and flux forecast model, where $C$ represents $CO_2$ concentration from CMAQ simulation and $E$ represents the $CO_2$ flux from the $CO_2$ flux forecast model. We have modified the relevant parts of the revised manuscript (Line 257−261).

*Line 265: model grid -> model grid point*

Yes, we have modified "model grid" as "model grid point", and we have checked the full text with the incorrect expression.

*Line 271-274: It is not well justified that data with |o-b|>5 ppm should be removed. How the choice of the threshold affects the inversion results?*

OMB (i.e., *o-b*, observation-minus-background) quality control method is based on the observation increments, which is used to check the background fields and adopted by many assimilation systems. In the EnKS assimilation scheme, the analysis $x^a$ is obtained by adding the innovations to the model forecast with weights $K$ (i.e. Kalman gain matrix), that are determined based on the estimated statistical error covariance of the forecast and the observations. Particularly, $K$ is obtained by minimizing the analysis error covariance with evolved forecast error covariance over time. In this study, the records with absolute biases (i.e., $|o-b|$) between the observation and background simulations greater than 5 ppm were removed, which are considered to have a lack of regional representativeness (Peng et al., 2023). Due to the spatial resolution ($64 \times 64$km) of our model, we cannot reproduce such observations. Moreover, the scenario of $|o-b| > 5.00$ was mostly found near the boundary of the model domain.

We have modified the relevant parts of the revised manuscript (Line 289−294).

*Line 281: Non-assimilated observations cannot be regarded as independent verification data. The filtering criteria (1) and (3) are the same as that for assimilated observations, and I don't quite get what the criteria (2) is about.*

Thank the reviewer for the comment. The assimilated and non-assimilated GOSAT $XCO_2$ observations are selected by different process of sifting (Table R2). These two sets of observations both used $XCO_2$ with "outcome_flag = 1" and precluded absolute biases between the observation and simulations greater than 5 ppm. Nevertheless, the main difference lies in step 2. The $XCO_2$ with the minimum "xco2_uncert" in the same model grid point at the same hour were used to assimilate, and other $XCO_2$ were used to validate.

Although some GOSAT observations which are not assimilated in the inversion were used to evaluate the posterior flux, they may have error distributions similar to those assimilated. Therefore, surface *in situ* observations from 14 sites are further used as independent observations to evaluate the inversion result in the revised manuscript. Comparison between surface observations, prior flux simulation, posterior flux simulation and the analysis for hourly $CO_2$ concentration is added in Table R1. According to the statistics listed in Table R1, the statistics of the analytical field (AN) are better than FC and CTRL, including RMSE and MAE, which gives a direct indication that the assimilation performs well. Taking improvement as example, the RMSE improvement rate between the FC and CTRL mostly ranges from –2.13% to 12.34% with an average of 2.48%, and the MAE improvement rate ranges from 0.08% to 9.73% with an average of 2.37%.

We have modified the relevant parts (Line 282−300), and Table 4 is further added and discussed in the revised manuscripts.

Table R2  GOSAT $XCO_2$ for assimilation and validation

|        | $XCO_2$ for assimilation | $XCO_2$ for validation |
|--------|--------------------------|-------------------------|
| Step 1 | Select $XCO_2$ with "outcome_flag = 1", | Select $XCO_2$ with "outcome_flag = 1", |
| Step 2 | Select $XCO_2$ with the minimum "xco2_uncert" in the same model grid point at the same hour | Select $XCO_2$ except for values minimum "xco2_uncert", in order to filter out all of the assimilated $XCO_2$ |
| Step 3 | Preclude record with absolute biases between the observation and simulations greater than 5 ppm | Preclude record with absolute biases between the observation and simulations greater than 5 ppm |

*Line 293: Natural fluxes are optimized/updated, not "assimilated"*

Yes, we have modified "assimilated" as "optimized", and we have checked the full text with the incorrect expression.

*Line 302-304: It is unclear whether boundary conditions are perturbed by 5% or 10%. More importantly, it is not justified whether 10% perturbation to natural fluxes is proper.*

The initial and boundary conditions are perturbed by adding Gaussian random noise with a standard deviation of 5%.

After completing the "forecast step" with the flux forecast model and CMAQ, Kalman gain matrix $K$ is obtained by minimizing the analysis error covariance with evolved forecast error covariance over time. Then, the associated analyzed state variables, $\boldsymbol{x}^a = \left[ \boldsymbol{C}^a, \boldsymbol{E}^a \right]^T$, can be updated by applying the EnKS constrained by GOSAT retrievals in the "analysis step". In addition, the distribution of ensemble spread of $CO_2$ flux in January 2016 is provided in Figure R1. It shows that the values of the ensemble spread ranges from 0.2 to 0.8 in most areas, which are consistent with our previous studies (Peng et al., 2015 in Figure 11c and Peng et al., 2023).

We have modified the description of the inversion algorithm in Section 2.2 (Line 310−315).

*Line 312: The word "high-risk" may not be suitable here.*

Yes, we have modified the expression. This traditional approach was adopted as a compromise to assess whether the *a posteriori* fluxes would enable improvements in the fit to the observed $CO_2$ concentrations. Fit to the observed $CO_2$ concentrations was analyzed with posterior and prior flux simulation, respectively. The aims are to check that inversions actually improve the model fits to the observations, which is a basic diagnostic of atmospheric inversions.

We have modified the expressions in the revised manuscript (Line 323−327).

*Line 325-334. Discussion on data coverage here is not related to either what is before or after. I do not see the flow of logic here.*

Thank the reviewer for the comment. We have substantially revised the paper to focus on the contribution of this study to the field. Attentions have also been paid to logic flows.

*Line 351: It is stated that the detector on GOSAT is "more sensitive to near-surface CO2 changes", but I don't know what this is compared to. And I do not see how this statement add to the discussion above.*

The shortwave near-infrared detectors mounted on GOSAT have been testified as being more sensitive to near-surface $CO_2$ changes, which is compared to the thermal infrared detectors such as AIRS, the Atmospheric Infrared Sounder on NASA's Aqua satellite. Because this statement was not closely related to the contribution of this study, we have deleted this statement from the text, and we have checked the full text with the logic flows.

*Line 373: I do not find any solid analysis showing that the calculation is reasonable or effective, except for some vague descriptions and comparisons.*

We are sorry about the overstatement "the flux analysis increments are reasonably and effectively calculated". At present, Section 3.1 focused on describing the pattern of observation and analysis increments. In Section 3.5, fit to the observed $CO_2$ concentrations was analyzed with posterior and prior flux simulation, respectively. The aims are to check that inversions actually improve the model fits to the observations, which is a basic diagnostic of atmospheric inversions. This statement has been deleted from the text and we have checked the full text with the overstatements and logic flows.

*Line 413-414: Logically, agreement with previous estimates does not provide a strong indication that your model transport is reliable.*

Thank the reviewer for the comment. Logically, agreement with previous estimates does not provide a strong indication that our model transport is reliable. However, the comparison aims at producing a collective assessment of the net carbon flux between the terrestrial ecosystems and the atmosphere in China. It aims in particular at investigating the capacity of the inversions to deliver consistent flux estimates at the country scale. The inversion systems differ by the transport model, the inversion approach, the choice of observation and prior constraints, enabling us to facilitate the international comparison and mutual recognition.

We have modified the relevant parts in the revised manuscript (Line 360−390).

*Line 471-472: I do not find results to support this claim.*

We are sorry about the overstatement "This indicates that the regional carbon assimilation system is calibrated well and performs reliably". This statement has been deleted from the text and we have

checked the full text with the overstatements.

*Line 489: Any evidence shows that a smaller daily variation is "more realistic"? I doubt whether the GOSAT data are sufficient to constrain the day-to-day variation given missing data and sparse sampling?*

We are sorry about the overstatement "which appears more realistic than that of the *a priori* estimates". First of all, the smoothing window of the flux forecast model was set as 4 days (Equation 1). This implies that not only useful observational information from the previous assimilation cycle has been made beneficial to the next assimilation cycle, but also the background error covariance matrix of our inversion system is flow dependent. Furthermore, observation at the current time has been designed to update fluxes from the previous 24 hours through EnKS assimilation scheme. Therefore, the assimilation system can fully absorb the existing observational information and optimize the prior flux to some extent.

At present, Section 3.3 focused on describing the regional characteristics of posterior fluxes. In Section 3.5, fit to the observed $CO_2$ concentrations was analyzed with posterior and prior flux simulation, which is a basic diagnostic of atmospheric inversions. Inversions actually improve the model fits to the hourly and daily observations (except for two sites with weekly observation).

This statement has been deleted from the text and we have checked the full text with the overstatements and logic flows.

The size of the biosphere carbon sink in China amounted to $-0.47$ PgC yr$^{-1}$ from JDAS with GOSAT constraints in this study. We have modified the relevant parts in the revised manuscript (Line 370−375).

*Section 3.3. The author found downward correction over forest and grassland and upward correction for cropland areas. This is an interesting finding, but no further information is presented.*

Thank the reviewer for the comment. Generally, the *a priori* biosphere fluxes are overestimated (~0.1– 0.3 μmole m$^{-2}$ s$^{-1}$) in the north (dominated by forest, grassland and cropland) and south (dominated by

forest and grassland) of China, while they are underestimated (~0.1–0.5 μmole m$^{-2}$ s$^{-1}$) primarily in central China where there is a large area of cropland (He et al., 2022). Fit to the observed $CO_2$ concentrations was analyzed with posterior and prior flux simulation, which is a basic diagnostic of atmospheric inversions. Inversions actually improve the model fits to the surface observations in forest areas (in Northeast, East and Southeast China), cropland areas (in North China), grassland areas (in Mongolia), Ocean (in Korea and Japan) and coastal areas (in Korea). Thus, downward correction over forest and grassland and upward correction for cropland areas has been evaluated against independent data.

We have modified the relevant parts in the revised manuscript with further information presented (Line 550−615).

---

## Author Comment (AC2)

**Response to Reviewer #2**

We thank the reviewer#2 for the insightful and detailed comments and suggestions, which helped to significantly improve the manuscript. The reviewer's comments are shown in *blue italics* with the author responses in black.

*General comments:*

*This study introduces the top down inversion of the natural biosphere carbon fluxes over China with a high horizontal resolution of about 64 km by joint optimization of initial $CO_2$ condition and biosphere carbon fluxes using GOSAT satellite observations. The magnitude of the estimated annual biosphere sink in China was consistent with most previous studies. In addition, the provincial biosphere carbon flux over China was also reestimated. Generally speaking, the paper is well written and scientific sound.*

*Main comments:*

- *It is unclear how the uncertainties of the background carbon fluxes are used in the data assimilation. Since the uncertainties of the background carbon fluxes are critical for the inversion, please clarify it more detail.*

Thank the reviewer for the comment. In CMAQ simulation, the prior prescribed $CO_2$ emissions come from both anthropogenic sources and natural sources, including fossil-fuel emission, terrestrial ecosystem flux, oceanic flux, and biomass burning emissions. In the assimilation, the natural flux (i.e., biosphere–atmosphere exchange and ocean–atmosphere exchange) were assimilated, while the fossil-fuel and biomass-burning fluxes were fixed based on bottom-up estimates, which follows previous inversion work and reflects our faith in inventory-based emissions for fossil fuels (Peters et al., 2007, 2010; Tian et al., 2014; Wang et al., 2019; Wang et al., 2020).

Considering the high level of uncertainty in simulated bioflux in current terrestrial biosphere models, those *a priori* biospheric fluxes were interpolated from the widely recognized CT2019B products, which is a global inverse model of atmospheric $CO_2$ to produce quantitative estimates of atmospheric carbon uptake and release. $CO_2$ fluxes $F(x, y, t)$ in CT2019B are parameterized according to

$$F(x,y,t) = \lambda(x,y,t)\big(F_{bio}(x,y,t) + F_{ocean}(x,y,t)\big) + F_{ff}(x,y,t) + F_{fire}(x,y,t)$$

where $F_{bio}, F_{ocean}, F_{ff}$, and $F_{fire}$ are prior flux model predictions for land biosphere, ocean, fossil fuel and biomass burning emissions respectively, and $\lambda$ represents a set of unknown multiplicative scaling factors applied to the fluxes, to be estimated in the assimilation. These scaling factors are the final product of CT2019B optimized fluxes.

In CarbonTracker, the flux dynamical model is applied to the ensemble-mean parameter values $\lambda$ as:

$$\lambda(t) = (\lambda_0 + \lambda(t-1) + \lambda(t-2))/3$$

where $\lambda(t)$ is the prior value of the scaling factors for timestep $t$, $\lambda_0$ is the initial prior vector with all elements set to 1.0, and $\lambda(t-1)$ and $\lambda(t-2)$ refers to the posterior scaling factors for the timestep $t-1$ and $t-2$ repectively. This model describes that parameter values $\lambda$ for a new time step are chosen as a combination of optimized values from the two previous time steps and a fixed overall prior value of 1.0.

In this study, the Equation (1) describes the flux forecast model in JDAS by taking the a priori flux, the analysis flux from the previous assimilation cycle, and the forecast concentration as independent variables. We can see that $M$ is used for linking the assimilated fluxes from the previous assimilating cycle, and $M$ was set to 3 in CarbonTracker. In JDAS real practice, $M$ was set to 4 days at the same time on each day to represent the average state of the biospheric diurnal variation at a certain seasonality level, as a result of several sensitivity tests which are not present here.

Measured $CO_2$ concentrations are the result of upstream surface fluxes and atmospheric transport process. Generally speaking, the longer in the past a flux event occurred, the smaller its impact will be on a given sample of air. Therefore, we choose an "assimilation window" to represents how far back in time we expect to be able to pinpoint a given flux signal from available measurements. CT2019B have designed the assimilation window length as 12 weeks. This helps to resolve fluxes in regions of the world with less dense observational coverage.

Similar to CarbonTracker which uses transport model as a forward operator in an ensemble fixed-lag Kalman smoother, JDAS is also extended to incorporate the ensemble Kalman smoother (EnKS)

feature along with EnSRF. The EnKS allows for a sequential processing of the measurements in time and is used to assimilate the concentrations and update the fluxes. Thus, EnKS that can take into future observations into account is used to assimilate the concentrations and update the fluxes. The smoothing window of EnKS (i.e. denoted as assimilation window hereafter) was set to 24 h in this study. In an assimilation cycle, the fluxes for the 24-h smoothing window have been designed to be optimized hour by hour successively.

The distribution of ensemble spread of $CO_2$ flux in January 2016 is provided in Figure R1. It shows that the values of the ensemble spread ranges from 0.2 to 0.8 in most areas, which are consistent with our previous studies (Peng et al., 2015 in Figure 11c and Peng et al., 2023).

We are sorry there is some confusion about the smoothing window in flux forecast model and EnKS in this manuscript. To avoid the confusion, we have modified the relevant parts in Section 2.1.2 (forecast model of ensemble fluxes) and Section 2.2.2 (EnKS assimilation scheme) in the revised manuscript.

[Figure]

Figure R1. The ensemble spread of $\lambda_{i,t}^a$ at model level 1 in January 2016, when $\beta$=80.

- *How the uncertainties of the boundary concentrations are considered in the study? How often the boundary and initial concentrations are imported from the CT2019B, and are the boundary concentrations are also optimized?*

Thank the reviewer for the comment. As the initial and lateral boundary atmospheric $CO_2$ concentrations, the global 4D $CO_2$ data were created using the optimized surface fluxes and simulated atmospheric transport of CarbonTracker, version CT2019B, from the National Oceanic and

Atmospheric Administration (NOAA), with a spatial resolution of $3° \times 2°$, 25 vertical levels, and a temporal resolution of 3 h, which represent the optimum estimate of the distribution of atmospheric $CO_2$ (Jacobson et al., 2020).

In each EnKS analysis step, CMAQ integrated and generated a 3D $CO_2$ concentration ensemble derived by the $N$ ensemble fluxes with perturbed $CO_2$ initial and boundary conditions. The ensemble assimilation was performed for the period 0000 UTC 25 December 2015 to 2300 UTC 31 December 2016 using the perturbed initial conditions and boundary conditions by adding Gaussian random noise with a standard deviation of 5%. In an assimilation cycle, the fluxes for the 24-h assimilation window have been designed to be optimized hour by hour successively. Accordingly, the fluxes have been adjusted 24 times before generating posterior fluxes. In this way, both the initial and boundary concentrations are optimized every hour.

We have modified the relevant parts in the revised manuscripts (Line 298−329). The detailed description of EnKS-based assimilation system configuration can be referred to Section 2.2 (JDAS $CO_2$ assimilation framework) and Section 2.4 (Experimental design and evaluation method) in the manuscript.

- *The a priori fluxes from CT2019B are at a 3-h intervals, how was the hour-by-hour assimilation conducted? Are the initial conditions are also optimized every hour?*

In an assimilation cycle, the fluxes for the 24-h assimilation window have been designed to be optimized hour by hour successively. Accordingly, the fluxes have been adjusted 24 times before generating posterior fluxes. Actually, the NOAA operational EnKF system, which is an EnSRF and modified with the EnKS feature, is further extended to jointly assimilate the $CO_2$ initial conditions and fluxes to update the flux and concentration fields, respectively. The EnKS allows for a sequential processing of the measurements in time, which updates the ensemble at prior times every time new observations are available. Thus, EnKS that can take into future observations into account is used to assimilate the concentrations and update the fluxes.

In this study, the state vector $\mathbf{x}$ includes the mass concentration $\mathbf{C}$ and the emission $\boldsymbol{E}$, i.e.

$\mathbf{x} = [\boldsymbol{C}, \boldsymbol{E}]^{\mathrm{T}}$. Here, the state variables of mass concentration $\mathbf{C}$ are the $CO_2$ concentrations. The ensemble forecast concentration fields of $CO_2$ are respectively used in calculating ensemble fluxes $\boldsymbol{E}_{i,t}^{f}$ as described in Section 2.2.1. The ensemble members of chemical fields $\mathbf{C}^f$ are forecasted using CMAQ, forced by the forecast emissions $\mathbf{E}^f$ whose initial conditions are previously analyzed concentration fields. Now, the background of the joint vector, $\mathbf{x}^f = \left[\mathbf{C}^f, \mathbf{E}^f\right]^{\mathrm{T}}$, has been produced. Then, the analyzed state vector, $\mathbf{x}^a = [\mathbf{C}^a, \boldsymbol{E}^a]^{\mathrm{T}}$, is optimized by applying the EnKS, respectively. The configurations of the EnKS were as follows: 1) ensemble size was set to 50; 2) the horizontal localization radius was 1280 km; 3) the covariance inflation factor β was set to 80; 4) the smoothing window (i.e. denoted as assimilation window hereafter) was set to 24 h, as sensitivity experiments about smoothing windows has been tested to find the optimum length in our previous study (Peng, et al., 2023). In addition, hour-by-hour assimilation was adopted attribute to the novel flux forecast model, fine-scale CMAQ forward hourly simulation output, as well as the hourly observations. Thus, the initial condition, boundary concentrations and flux are optimized every hour.

We have modified the relevant parts (Section 2.2) in the revised manuscripts, and we hope we can make the meaning clear now.

- *It is better to separate the results and discussion.*

Thank the review for the comment. And we have separated the results and discussion in the revised manuscript (Section 3 and Section 4).

*Specific comments:*

- *P9 How do you determine the values of the horizontal covariance localization radius and the inflation factor?*

The localization radius 1280 km follows our previous research including Peng et al., 2015, Peng et al., 2018, Peng et al., 2023, which localize the impact of observation and ameliorate spurious error correlations between observations and state variables. Thus, covariance localization (Houtekamer & Mitchell, 2001) with the Gaspari and Cohn (Gaspari & Cohn, 1999) function of 1280 km length scale, are utilized.

Moreover, the covariance inflation factor $\beta$ was set to 80 to preserve the ensemble spread ranging to some extent. The distribution of ensemble spread of $CO_2$ flux in January 2016 is provided in Figure R1. It shows that the values of the ensemble spread ranges from 0.2 to 0.8 in most areas, which are consistent with our previous studies (Peng et al., 2015 in Figure 11c and Peng et al. 2023).

We have modified the relevant parts in the revised manuscript (Line 265–270).

Here are the above-mentioned references.

Peng, Z., Zhang, M. G., Kou, X. X., Tian, X. J., & Ma, X. G. (2015). A regional carbon flux data assimilation system and its preliminary evaluation in East Asia. *Atmospheric Chemistry and Physics*, 15, 1087–1104. https://doi.org/10.5194/acp-15-1087-2015.

Peng, Z., Lei, L. L., Liu, Z. Q., Sun, J. N., Ding, A, J., Ban, J. M., et al. (2018). The impact of multi-species surface chemical observation assimilation on air quality forecasts in China. *Atmospheric Chemistry and Physics*, 18, 17387–17404. https://doi.org/10.5194/acp-18-17387-2018

Peng, Z., Kou, X. X., Zhang, M. G., Lei, L. L., Miao, S. G., Wang, H. M., Jiang, F., Han, X., and Fang, S. X. (2023). $CO_2$ flux inversion with a regional joint data assimilation system based on CMAQ, EnKS, and surface observations. *Journal of Geophysical Research-Atmosphere*, 128, e2022JD037154. https://doi. org/10.1029/2022JD037154

Houtekamer, P. L., & Mitchell, H. L. (2001). A sequential ensemble Kalman filter for atmospheric data assimilation. *Monthly Weather Review*, 129, 123–137. https://doi.org/10.1175/1520-0493(2001)129<0123:ASEKFF>2.0.CO;2

Gaspari, G., & Cohn S. E. (1999). Construction of correlation functions in two and three dimensions. *Quarterly Journal of the Royal Meteorological Society*, 125, 723–757. https://doi.org/10.1002/qj.49712555417

▪ *Why the Table 2 is firstly appeared in the main text?*

Thank the review for the comment. And we have adjusted the order of the tables.

*Line 526 The horizontal resolution of the CMAQ model in the study is about 64 km, why the results cannot resolve the Shanghai?*

The total area of Shanghai is 6340.5 km$^2$. The CMAQ configuration used here was 64 $\times$ 64 km$^2$ (i.e. 4096 km$^2$ each grid) fixed grid cells centered at 35 N and 116 E in a rotated polar stereographic map projection. This domain, having 105 (west–east) $\times$ 86 (south–north) grid points, covered the whole of mainland China and its surrounding regions (Fig. 1). Thus, owing to the insufficient grid resolution, Shang has been mixed with the neighbouring areas, especially Jiangsu and Zhejiang provinces. In addition, Hong Kong and Macao are not discussed, because the results cannot resolve these areas too.

---

## Author Response (AR2)

**Response to Reviewer #1**

We thank the reviewer#1 for the insightful and detailed comments and suggestions, which helped to significantly improve the manuscript. The reviewer's comments are shown in *blue italics* with the author responses in black.

*General comments:*

*Kou et al. conducted a regional atmospheric inversion analysis using GOSAT satellite $XCO_2$ products to constrain yearly $CO_2$ net fluxes in China. To quantify China's net ecosystem-atmosphere $CO_2$ exchange, they utilized the CMAQ-ENKS system. The regional inversion study is crucial in complementing commonly-used global inversion frameworks to accurately diagnose $CO_2$ NEE over complex environments. The manuscript is well-written and includes an in-depth discussion. I recommend it for publication in ACP pending the authors' response to the following comments.*

*1. After data screening, the size of GOSAT $XCO_2$ data may not be sufficient to inversely constrain all yearly $64 \times 64km^2$ and hourly $CO_2$ fluxes in China. How did you reconcile the observational limit and the specific spatial-temporal flux state vector? How did you determine the posterior uncertainty associated with the posterior $CO_2$ flux estimation, and did the posterior uncertainty involve the uncertainty due to the unbalance between the numbers of obs and the specific resolutions applied here? Please explain and clarify these questions in the response and main manuscript.*

Thank the reviewer for the comment. (1) The update for $CO_2$ flux is given by the observation innovation and the correlations between $CO_2$ concentrations and emissions, while the correlations are naturally provided by the physics- and dynamics-based numerical model. (2) Although there are limited observation numbers, the observations are available of 1 hour. Thus through hourly update along with hourly model advances, the spatially sparse observations can sufficiently constrain the $CO_2$ flux, which can be demonstrated by the results. (3) Given the EnKF algorithm, the posterior uncertainty is proportional to the prior uncertainty but with a smaller magnitude. Based on hourly update, the posterior uncertainty contains the same flow-dependent information as the prior uncertainty. (4) For both chemistry assimilation and numerical weather prediction, it is commonly that the dimension of observation is much smaller than the dimension of state vector. Thus data assimilation helps to use the limited observations to constrain the state vector.

We have modified the relevant parts in the revised manuscript (Line 292–301); please check if it is clear now.

*2. Line 316: How do you justify that the 7-day spin-up is enough to construct the inversion estimation, given that the domain is relatively large? Please clarify this point.*

Thank the reviewer for the comment. The 7-day spin-up has been testified by a series of OSSEs (Observing System Simulation Experiments) in Peng et al., 2015, which conducted pseudo-satellite-observation and CMAQ assimilation with the same model domain and horizontal resolution (i.e. 64km× 64km over East Asia). Over the first few days, the assimilated $CO_2$ diverged from the modeled fields, generally moving closer to the observations, indicating that a spin-up time of about 7 days is required for the assimilation system to respond. In addition, the spin-up time for different assimilation systems implies that the long lifetime of atmospheric $CO_2$ and limited number of observations need to be taken into account in the regional joint assimilation framework (Tian et al., 2014, Peng et al., 2015, Kou et al., 2017).

We have modified the relevant parts in the revised manuscript (Line 334–337).

Here are the above-mentioned references.

Kou, X. X., Tian, X. J., Zhang, M. G., Peng, Z., & Zhang, X. L. (2017). Accounting for $CO_2$ variability over East Asia with a regional joint inversion system and its preliminary evaluation. *Journal of Meteorological Research*, 31(5), 834–851. https://doi.org/10.1007/s13351-017-6149-8.

Peng, Z., Zhang, M. G., Kou, X. X., Tian, X. J., & Ma, X. G. (2015). A regional carbon flux data assimilation system and its preliminary evaluation in East Asia. *Atmospheric Chemistry and Physics*, 15, 1087–1104. https://doi.org/10.5194/acp-15-1087-2015.

Tian, X., Xie, Z., Liu, Y., Cai, Z., Fu, Y., Zhang, H., & Feng, L. (2014) A joint data assimilation system (Tan-Tracker) to simultaneously estimate surface $CO_2$ fluxes and 3-D atmospheric $CO_2$ concentrations from observations. *Atmospheric Chemistry and Physics*, 14, 13281–13293. https://doi.org/ doi:10.5194/acp-14-13281-2014, 2014.

*3. The Results section needs to be modified to be concise. It's better to move some content in Section 3 (Results) to Section 4 (Discussion). Please focus on your estimates for Section 3, modify and condense the previous studies (Lines 363-376) to either Discussion or Introduction sections.*

Thank the reviewer for the comment. We have modified the Results section to be concise, and focused on our estimates for Section 3. Line 363–376 has moved to Discussion section.

We have modified the relevant parts in the revised manuscript (Line 381–392, Line 408–411, Line 538–539 and Line 549–561).

*4. How does the performance of the posterior $CO_2$ flux estimates for the ocean area of the domain compare? Does GOSAT have the same algorithm to handle $XCO_2$ over land and ocean? Did you use the same QA/QC to determine your assimilated $XCO_2$ data for land and ocean?*

Thank the reviewer for the comment. In this study, GOSAT $XCO_2$ retrievals were from NASA's ACOS_L2_Lite_FP.9r (data available at https://oco2.gesdisc.eosdis.nasa.gov/data/GOSAT_TANSO_Level2/), this version of processing supports both nadir and glint soundings. In the case of soundings over water, a check was made to ensure the observation was made in glint mode in ACOS retrievals.

In the present study, the GOSAT $XCO_2$ were introduced in the EnKS-based assimilation framework to constrain China's biosphere sink. The CMAQ-simulated $CO_2$ concentrations profiles were mapped into the GOSAT satellite retrieval levels and then vertically integrated based on the satellite averaging kernel according to the following equation:

$$XCO_2^f = XCO_2^p + \sum_{k=1}^{N_{lev}} \left\{ \left[ \left( y_k^f - y_k^p \right) A_k \right] h_k (1-w)^{-1} \right\} \tag{S1}$$

Then state variables can be updated by applying the EnKS constrained by GOSAT retrievals over land and ocean in the analysis step (Equation S2).

$$x^a = x^f + K(y - H(x^f)) \tag{S2}$$

Details of the Equation S1 and S2 are provided in Section 2.2.2.

Before being applied in assimilation, we use the same data screening strategy to handle $XCO_2$ over land and ocean. The retrievals for the glint soundings over oceans have relatively larger uncertainty, and thus many data over oceans are excluded in our inversions in terms of data screening strategy (Figure 2). On the other hand, our present study focus on top-down estimation of China's biosphere sink, fully investigation of posterior $CO_2$ flux estimates for the ocean area is outside the scope of this work and is therefore not discussed any further here. In the future, we'll further study China's ocean carbon source and sinks based on satellite retrievals and regional CTM assimilation in depth.

We have modified the relevant parts in the revised manuscript (Line 274–278, Line 292–301, and Line 673–676).

*5. Line 344-345: "This discrepancy of the seasonal scale..." This sentence is not clear to me. Did you mean the mixing between biospheric and fossil-fuel sources or the differences between them?*

Thank the reviewer for the comment. Table 1 indicates that the point-by-point uncertainty is larger in summer and lower in spring and autumn. The difference in seasonal performance could be partly due to the uncertainties in the spatial and temporal variations of the biosphere flux estimation and fossil-fuel inventories.

We have modified the relevant parts in the revised manuscript (Line 361–364); please check if it is clear now.

*6. Line 341: "(1.99 and 2.41..)" lacks a unit.*

Thank the reviewer for the comment. The unit for MAE and RMSE is ppm. We have added unit (i.e. ppm) in the revised manuscript (Line 360).

*7. Was the system designed to prevent unrealistic and non-physical negative flux estimates due to Gaussian assumption/perturbation? Please clarify this point.*

Thank the reviewer for the comment. In anthropogenic emission assimilation, negative flux estimates are unrealistic and non-physical, so they are eliminated. This might result in the Gaussian assumption not being satisfied. However, in carbon data assimilation, negative flux refers to the uptake of atmospheric

$CO_2$ by photosynthesis exceeds $CO_2$ released by respiration, especially in the growing season. Negative flux in carbon assimilation is realistic and reasonable, which are not excluded. In this way, Gaussian assumption is satisfied in JDAS carbon assimilation.

We have modified the relevant parts in the revised manuscript (Line 219–221).

*8. Section 2.3, the last two paragraphs (Lines 284-302) have redundant information in terms of the $XCO_2$ data QA/QC (3-step screening strategy). Please modify them to be more concise.*

Thank the reviewer for the comment. The assimilated and non-assimilated GOSAT $XCO_2$ observations are selected by different process of sifting (Table R1). These two sets of observations both used $XCO_2$ with "outcome_flag = 1" and precluded absolute biases between the observation and simulations greater than 5 ppm. Nevertheless, the main difference lies in step 2. The $XCO_2$ with the minimum "xco2_uncert" in the same model grid point at the same hour were used to assimilate, and other $XCO_2$ were used to validate.

Table R1    GOSAT $XCO_2$ for assimilation and validation

|        | $XCO_2$ for assimilation | $XCO_2$ for validation |
|--------|--------------------------|------------------------|
| Step 1 | Select $XCO_2$ with "outcome_flag = 1", | Select $XCO_2$ with "outcome_flag = 1", |
| Step 2 | Select $XCO_2$ with the minimum "xco2_uncert" in the same model grid point at the same hour | Select $XCO_2$ except for values minimum "xco2_uncert", in order to filter out all of the assimilated $XCO_2$ |
| Step 3 | Preclude record with absolute biases between the observation and simulations greater than 5 ppm | Preclude record with absolute biases between the observation and simulations greater than 5 ppm |

We have modified the relevant parts in the revised manuscript (Line 303−320).

*9. The posterior uncertainty is not presented in the study. It would be expected that this information is displayed for Figures 3, 4, and 6. Echoing my previous comment, please add or clarify the posterior uncertainty considerations/treatment in the study.*

Thank the reviewer for the comment. Similar to CarbonTracker which uses transport model as a forward operator in an ensemble fixed-lag Kalman smoother, JDAS is also extended to incorporate the EnKS

feature. The EnKS allows for a sequential processing of the measurements in time and is used to assimilate the concentrations and update the fluxes. Thus, EnKS that can take into future observations into account is used to assimilate the concentrations and update the fluxes. The smoothing window of EnKS (i.e. denoted as assimilation window hereafter) was set to 24 h in this study. In an assimilation cycle, the fluxes for the 24-h smoothing window have been designed to be optimized hour by hour successively.

In the joint assimilation framework, besides the application of CMAQ to generate ensemble $CO_2$ concentrations, a flux forecast model was also designed to represents flux variations on account of fluxes acting as model forcing. Consequently, after completing the "forecast step", Kalman gain matrix $K$ is obtained by minimizing the analysis error covariance with evolved forecast error covariance over time. Then, the associated analyzed state variables, $\boldsymbol{x}^a = \left[ \boldsymbol{C}^a, \boldsymbol{E}^a \right]^T$, can be updated by applying the EnKS constrained by GOSAT retrievals in the "analysis step". Furthermore, the distribution of ensemble spread of $CO_2$ flux in January 2016 is provided in Figure R1. It shows that the values of the ensemble spread ranges from 0.2 to 0.8 in most areas, which are consistent with our previous studies (Peng et al., 2015 in Figure 11c and Peng et al., 2023).

[Figure]

Figure R1. The ensemble spread of $\lambda_{i,t}^a$ at model level 1 in January 2016, when $\beta$=80.

Here are the above-mentioned references.

Peng, Z., Zhang, M. G., Kou, X. X., Tian, X. J., & Ma, X. G. (2015). A regional carbon flux data assimilation system and its preliminary evaluation in East Asia. *Atmospheric Chemistry and Physics*,

15, 1087–1104. https://doi.org/10.5194/acp-15-1087-2015.

Peng, Z., Kou, X. X., Zhang, M. G., Lei, L. L., Miao, S. G., Wang, H. M., Jiang, F., Han, X., and Fang, S. X. (2023). CO$_2$ flux inversion with a regional joint data assimilation system based on CMAQ, EnKS, and surface observations. *Journal of Geophysical Research-Atmosphere*, 128, e2022JD037154. https://doi. org/10.1029/2022JD037154

We have modified the relevant parts in the revised manuscripts (Line 271−278), and Figure 1 is further added and discussed.

*10. Analysis increments are not clear to me. How did you calculate them? Did you use FC minus CTRL? The adjustments between FC and CTRL are very small over the ocean.*

Thank the reviewer for the comment. (1) The analysis-minus-background, AMB, (i.e., $\boldsymbol{x}^a - \boldsymbol{x}^{\mathrm{b}}$) is denoted as "analysis increments". Fig. 2 focus on the discussion of flux analysis increments (i.e., $\boldsymbol{E}^a - \boldsymbol{E}^{\mathrm{b}}$) to certify that GOSAT XCO$_2$ is effectively absorbed in JDAS. (2) FC and CTRL experiments were further designed to assess the quality of the inversion results. One set of experiments was forced by the optimized *a posteriori* fluxes (denoted as FC), and the other was forced by the prescribed *a priori* fluxes as a control experiment (denoted as CTRL). This traditional approach was adopted as a compromise to assess whether the *a posteriori* fluxes would enable improvements in the fit to observed CO$_2$ concentrations, including non-assimilated GOSAT as well as surface observations from 14 sites. (3) The retrievals for the glint soundings over oceans have relatively larger uncertainty, and thus many data over oceans are excluded in our inversions in terms of data screening strategy (Figure 2). In consequence, the adjustments between FC and CTRL are very small over the ocean.

We have modified the relevant parts in the revised manuscripts (Line 366−368).

*11. Figure 1: The colors for the analysis increment plots are too light. Please modify the color bar range to have a better display of the contrast between sink and source.*

Thank the reviewer for the comment. We have modified the color bar range (Fig. R2) to have a better display of the contrast between increases and decreases in the revised manuscript (Fig. 2); please check if it is clear now.

[Figure]

**Figure R2**. Observation increments (XCO$_2$; unit: ppm) and analysis increments (biosphere flux; unit: μmole m$^{-2}$ s$^{-1}$) in (a, b) January, (c, d) July, and (e, f) the whole year of 2016.